# Deep Mutational Scanning of FDX1 Identifies Key Structural Determinants of Lipoylation and Cuproptosis

Jeffrey C. Hsiao[1,8], Douglas M. Warui[2,3,8], Jason J. Kwon[4], Margaret B. Dreishpoon[1], Nolan R. Bick [4], Tamra Blue-Lahom [2,3], David E. Root [4], Squire J. Booker [2,3,5,6,7] & Peter Tsvetkov [1,4] ✉

Cuproptosis is a recently described form of regulated cell death triggered by ionophore-induced copper (Cu) overload in mitochondria. It is critically dependent on ferredoxin 1 (FDX1), a mitochondrial iron-sulfur cluster containing protein that acts as an electron shuttle. FDX1 reduces ionophore-bound Cu(II) to Cu(I), thereby triggering its release, and promotes mitochondrial protein lipoylation, which is directly targeted by the released copper to drive cell death. Despite the pivotal role of FDX1 in cuproptosis, the structural determinants underlying its distinct functions remain unclear. To address this, we performed deep mutational scanning on FDX1 and find that two conserved solvent-exposed residues, D136 and D139, on alpha helix 3 are essential for both cuproptosis and lipoylation. Charge-reversal mutations at these positions abolish FDX1's ability to induce cuproptosis and support lipoylation in cells, despite retaining full enzymatic activity in vitro. Guided by structural and genomic analyses, we further identify dihydrolipoamide dehydrogenase (DLD), the E3 subunit of lipoylated complexes as an alternative FDX1 reductase both in cells and in vitro. Together, these findings establish the acidic alpha helix 3 of FDX1 as a critical interface for its upstream regulation and suggest that FDX1's roles in cuproptosis and in lipoylation are both structurally and functionally linked.

Cuproptosis is a recently identified form of regulated cell death triggered by the toxic accumulation of copper (Cu) in mitochondria, a process facilitated by Cu ionophores such as elesclomol (ES)[1,2]. Multiple CRISPR/Cas9 whole-genome screens and genetic analyses of hundreds of cancer cell lines have identified ferredoxin 1 (FDX1) as a key regulator of ES-induced cuproptosis. FDX1 promotes cuproptosis by directly reducing ES-bound Cu(II) to Cu(I), leading to Cu release, ES recycling, and subsequent Cu-induced cytotoxicity[1–3]. This toxicity arises from the selective targeting of the mitochondrially localized dihydrolipoyl transacetylase (DLAT) and iron-sulfur (Fe-S) cluster proteins, resulting in lipoylated protein aggregation and Fe-S cluster destabilization, respectively[2].

The crucial role of FDX1 in regulating cuproptosis highlighted the importance of establishing its natural role in human cells. In steroidogenic tissues, FDX1 regulates steroid hormone synthesis by donating electrons to various cytochrome P450

[1]Department of Pathology and Cancer Center, Beth Israel Deaconess Medical Center, Boston, MA, USA. [2]Department of Chemistry, The Pennsylvania State University, University Park, PA, USA. [3]Department of Chemistry, School of Arts and Sciences, University of Pennsylvania, Philadelphia, PA, USA. [4]Broad Institute of MIT and Harvard, Cambridge, MA, USA. [5]Department of Biochemistry & Molecular Biology, The Pennsylvania State University, University Park, PA, USA. [6]Howard Hughes Medical Institute, Chevy Chase, MD, USA. [7]Department of Biochemistry and Biophysics, Perelman School of Medicine at the University of Pennsylvania, Philadelphia, PA, USA. [8]These authors contributed equally: Jeffrey C. Hsiao, Douglas M. Warui. ✉e-mail: ptsvetko@bidmc.harvard.edu

(CYP) proteins such as CYP11A1[4–6]. Beyond steroidogenesis, FDX1 has been suggested to play a role in the biosynthesis of bile acid, heme, and vitamin A and D by donating electrons to other cytochrome P450 enzymes[7]. Over the years there has been some debate as to the spectrum of FDX1 downstream substrates. For instance, FDX1 was suggested to be involved in Fe-S cluster biogenesis[8,9]. However, in recent genetic and proteomic studies in human cells[10–12], this role was largely disproved, showing that the predominant non-steroidogenic role of FDX1 in human cells is in regulating protein lipoylation[2,10–12] and maintaining electron transport chain (ETC) Complex IV integrity[11,13]. The role of FDX1 in regulating lipoylation was also confirmed in vitro, showing that FDX1 is essential for lipoic acid synthase (LIAS) activity by promoting the reductive cleavage of S-adenosylmethionine (SAM) to a required 5'-deoxyadenosyl 5'-radical intermediate[10]. While some controversy remains regarding the full spectrum of endogenous, functional downstream substrates of human FDX1, there is consensus that its upstream electron donor is the mitochondrial NADPH-dependent ferredoxin reductase (FDXR)[7,14,15].

FDX1 plays a dual role in cuproptosis (Fig. 1A). First, FDX1 reduces the Cu(II) bound to ES, prompting its release in mitochondria[1,3]. Second, FDX1 regulates protein lipoylation, the downstream target of the released Cu. The absence of protein lipoylation is sufficient to rescue ES induced cuproptosis, establishing lipoylation as a requirement to facilitate cuproptosis through a toxic gain-of-function mechanism[2]. Thus, to mechanistically understand the regulatory roles of FDX1 in promoting cuproptosis, there is a need to first establish the structural elements of FDX1 that are required for its distinct functions.

In this study, we systematically define the structural determinants of FDX1 required for cuproptosis. Deep mutational scanning (DMS) identifies two conserved acidic residues (D136 and D139) on α-helix 3 as critical for both copper-induced cell death and protein lipoylation, demonstrating that these functions cannot be uncoupled. We further uncover dihydrolipoamide dehydrogenase (DLD) as an upstream activator of FDX1, revealing a distinct layer of regulation that mechanistically links the lipoylation machinery to cuproptosis.

## Results

### Deep mutational scanning defines FDX1 structure-function

To identify the structural elements of FDX1 required for cuproptosis induction, we performed a DMS analysis. First, we established two cell line models lung adenocarcinoma ABC1 and human embryonic kidney 293 cells T expressing SV40 large T-antigen (HEK293T) in which *FDX1* was deleted using electroporation of Cas9 ribonucleoprotein complexes, avoiding constitutive Cas9 overexpression. As expected, knockout (KO) of *FDX1* conferred resistance to ES-induced cuproptosis, whereas reconstitution of FDX1 in KO cells re-sensitized them, yielding even greater sensitivity than in the wild-type control (~100-fold increase) in the ABC1 cell line (Fig. 1B; Supplementary Fig. 1A). This strong dependence on FDX1 protein levels created an ideal system for a DMS analysis.

To systematically assess the functional importance of individual residues, we generated a comprehensive library of FDX1 mutants, where each amino acid was substituted with every possible residue, including stop codons. This library was introduced into *FDX1* KO ABC1 and HEK293T cells (Fig. 1C). Cells expressing FDX1 mutants were grown in the presence or absence of ES, and the ability of each mutant to restore ES sensitivity was assessed after 14 and 11 days in ABC1 and HEK293T cells, respectively. As expected, nonsense mutations failed to re-sensitize *FDX1* KO cells to ES, whereas silent mutations did (Fig. 1D), confirming the specificity of our approach and establishing the confined boundaries for missense mutation analysis.

### Energetic mapping reveals selective cuproptosis mutants

To quantify the impact of individual mutations, we measured the $\log_2$ fold-change (LFC) viability for each FDX1 mutant, centered them to the means of silent mutants and nonsense mutants (i.e., scaled LFC), and visualized these data using a heatmap for both ABC1 (Fig. 2A) and HEK293T (Supplementary Fig. 1B) cells. The effects of FDX1 mutations on ES sensitivity were highly correlated across the two lines (Supplementary Fig. 1J), and several clusters of positions with mutational intolerance emerged. First, we observed a cluster around residues 40 to 50 at the N terminus that is more pronounced in the ABC1 cell line (Fig. 2A). We hypothesized that this region contributes to mitochondrial

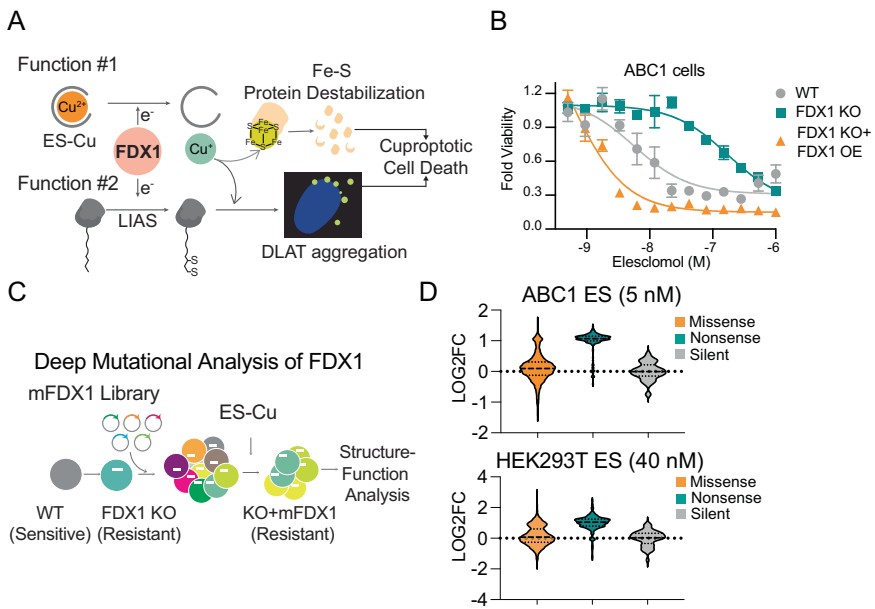

**Fig. 1 | Deep mutational scanning (DMS) of FDX1. A** Two functions of FDX1: (1) reducing elesclomol (ES)-bound Cu(II) to Cu(I), which facilitates its release and subsequent toxicity; and (2) enabling LIAS mediated lipoylation. **B** Viability of parental, FDX1 KO, or FDX1 KO with FDX1 reconstituted ABC1 cells 72 h after treatment with ES. Viability data is presented as mean ± SD of four biological replicates. Immunoblots of indicated cell lines are provided in Fig. 3D. **C** DMS screen experimental design. mFDX1 denotes mutant FDX1. **D** Normalized distributions of ABC1 or HEK293T cells harboring missense, nonsense, or silent mutations of FDX1. LOG2FC denotes $\log_2$ fold change in viability.

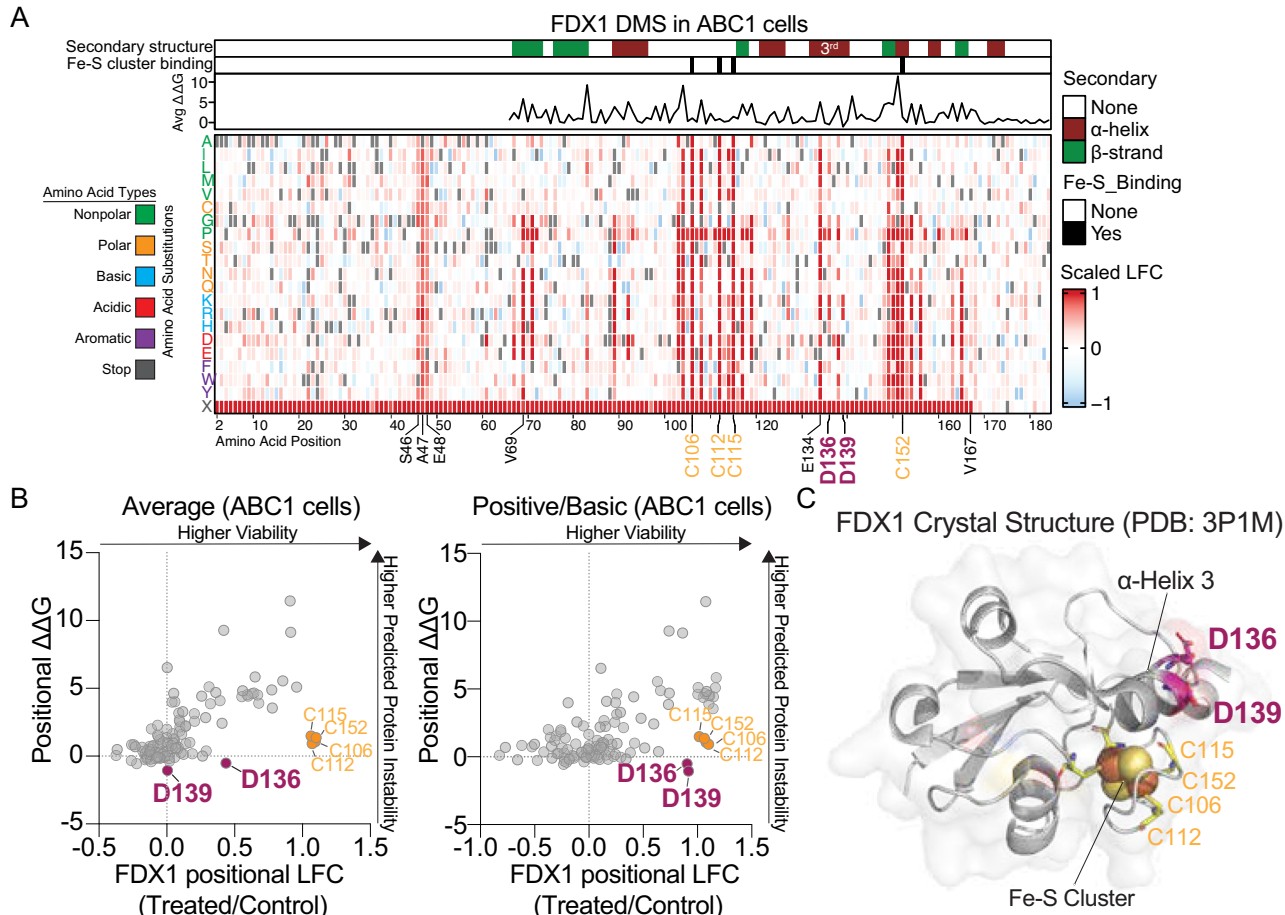

**Fig. 2 | DMS screen reveals D136 and D139 as key residues on FDX1 regulating cuproptosis. A** Heatmap of the results of the DMS screen in ABC1 cells. Colors are plotted as a function of the mean log$_2$ fold change (LFC) in viability of duplicates scaled relative to the mean viability scores obtained from the nonsense (amino acid X) and silent mutation distributions (Fig. 1D). The redder the color, the higher the viability. The gray bars indicate either no mutation (i.e., original amino acid) or missing data acquisition. The secondary structures of specific segments of FDX1 are indicated; the third alpha helix is emphasized. Residues within clusters of mutational intolerance are emphasized. Amino acids are colored coded by their chemical properties. The average ΔΔG, which predicts a mutation's impact on protein stability, is indicated at each position on FDX1. **B** The left scatterplot shows the average viability LFC and average positional ΔΔG for each FDX1 amino acid position when mutated to all possible residues. The right panel shows the same, but only when mutated to positive/basic residues (arginine, lysine, histidine). Indicated in orange are the Fe-S cluster binding cysteines; and in magenta, residues D136 and D139. **C** Crystal structure of human FDX1 PDB: 3P1M in complex with an Fe-S cluster shown in spheres. Residues D136 and D139 are presented in magenta sticks and the four cysteines that coordinate the Fe-S cluster in yellow sticks. Transparent surface model is shown to highlight solvent-exposure of D136 and D139.

targeting and that substitutions to this region would impair import and diminish FDX1 function (Supplementary Fig. 1C, D). Second, variants at V69 and F71 mapped to a hydrophobic β-strand core, and hydrophobic substitutions at these sites were largely tolerated (Supplementary Fig. 1E). Third, residue E134 on the third alpha helix (α-helix 3) coordinates the side chain of R149 and the backbone at L149, suggesting that its mutation disrupts the α-helix 3-stabilizing network and indicates an important structural role for this helix (Supplementary Fig. 1F). Finally, mutations of the iron-sulfur cluster coordinating cysteines produced strong loss-of-function phenotypes, as expected (Fig. 2C). It should also be noted that substitutions in the extreme C terminus (>V167) were largely tolerated, consistent with its predicted disorder and a region that is not essential for FDX1 function (Supplementary Fig. 1G). Together, these patterns indicate that mutational intolerance that attenuate FDX1-mediated cuproptosis arises from substitutions that impair mitochondrial targeting, destabilize protein structure, or disrupt cofactor binding.

We next sought to distinguish between two classes of FDX1 mutations that impair the ability to re-sensitize *FDX1* KO cells to ES: (1) missense mutations that disrupt FDX1 structure, leading to enzymatically inactive forms of FDX1, and (2) mutations that selectively impair FDX1's ability to regulate cuproptosis while preserving enzymatic

function. To differentiate between these classes, we performed an in silico FoldX mutational analysis[16,17] to predict the mean free-energy change (ΔΔG) for each position in FDX1. Loss-of-function (LOF) variants predicted to cause global protein instability exhibit high ΔΔG values. It should be noted that this metric represents a computational prediction rather than an experimental measurement and may therefore result in false positives/negatives[18]. Plotting the predicted ΔΔG against the scaled LFC in ES-induced viability (Fig. 2B and Supplementary Fig. 1H, left panels) revealed that most variations causing resistance were associated with high ΔΔG values, consistent with structural destabilization. Resistant variants that did not exhibit high ΔΔG values but did exhibit high scaled LFC values included the essential Fe-S cluster-binding cysteine residues (C106, C112, C115, C152), as the Fe-S cluster-binding properties of FDX1 were not accounted for in FoldX calculations. In addition to these expected findings, two residues—D136 and D139—stood out as critical regulators of ES-Cu sensitivity (Fig. 2C). These positions were susceptible to LOF upon mutation to positively charged residues, specifically lysine or arginine residues (Fig. 2B and Supplementary Fig. 1H, right panels; Supplementary Fig. 1I). The mutational intolerance to charge-specific substitutions at D136 and D139 while not impacting FDX1 stability

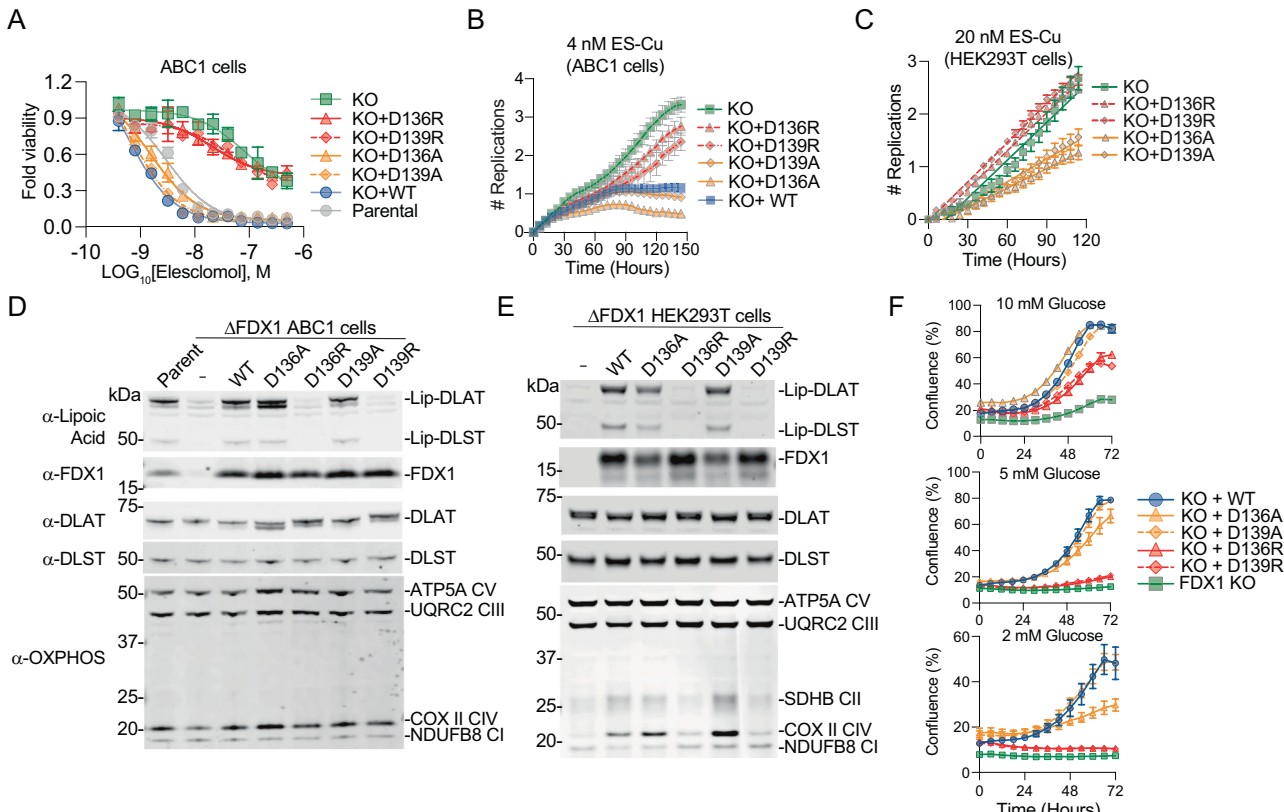

**Fig. 3 | Charge flip on residues D136 and D139 abolishes FDX1 mediated cuproptosis and protein lipoylation in cells. A** Viability of the indicated cell lines 72 h after treatment with ES. **B**, **C** Proliferation (represented as the number of replications) of the indicated cell lines were measured over the indicated times using live cell imaging. **D**, **E** Cell lysates obtained from the indicated cell lines were subjected to immunoblotting and assessed with the indicated antibodies shown on the left. **F** Proliferation (represented as the percentage of confluence) of the indicated HEK293T cell lines were measured over the indicated times in varying glucose conditions using live cell imaging. In (**A**–**C**) and (**F**), data is presented as mean ± SD of three (**A**), eight (**B**), eight (**C**), and three (**F**) biological replicates.

suggested that these solvent-exposed residues, which sit on FDX1's α-helix 3, play a role in mediating intermolecular, electrostatic interactions important for cuproptosis (Fig. 2C).

## Charge reversal on α-helix 3 disrupts FDX1 function

To validate our findings, we generated four D136 and D139 mutants: two alanine substitutions (D136A, D139A) to create neutral charge mutants that eliminate the negative charge while maintaining overall structure, and two arginine substitutions (D136R, D139R) to introduce a charge flip by replacing the negatively charged aspartic acid with a positively charged residue. These mutants, along with wild-type (WT) FDX1, were complemented to similar levels in FDX1 knockout (KO) ABC1 and HEK293T cells, and their sensitivity to ES or ES-Cu was assessed (Fig. 3A–C).

As expected, WT FDX1 fully restored ES/ES-Cu sensitivity in *FDX1* KO cells. Similarly, the neutral D136A and D139A mutants fully re-sensitized cells to ES/ES-Cu, demonstrating that the negative charge at these positions is not strictly required for FDX1 function. In contrast, cells expressing the charge-flipped D136R and D139R mutants failed to regain ES/ES-Cu sensitivity. These findings highlight that while residues D136 and D139 located on α-helix 3 (Fig. 2C) are not intrinsically essential (as some mutations are tolerated), their substitutions to positively charged residues specifically inhibit the ability of FDX1 to induce cuproptosis.

FDX1 facilitates cuproptosis by reducing Cu(II)-ES to Cu(I) and regulating protein lipoylation[1–3,10–12]. Therefore, we further assessed the impact of D136 and D139 substitutions on protein lipoylation levels in ABC1 and HEK293T cells. As expected, FDX1 deletion led to a complete loss of DLST and DLAT lipoylation (Fig. 3D, E). Complementation

of WT, D136A, or D139A mutants completely rescued lipoylation, suggesting that these aspartic acid residues are dispensable and can be substituted by other amino acids without an overall loss of enzymatic function. However, the charge-flipped D136R and D139R mutants completely abolished FDX1's ability to restore lipoylation (Fig. 3D, E). Previously, FDX1 was shown to regulate biogenesis of mitochondrial cytochrome c oxidase (COX) in human cells[13]. The effect of *FDX1* KO on COX subunit II (COX-II) levels seems to be cell line specific, with dramatic loss of COX-II in HEK293T but not ABC1 FDX1 KO cells (Fig. 3D, E). In HEK293T cells, expression of wild-type FDX1 as well as the D136A and D139A mutants successfully restored COX-II levels, whereas the D136R and D139R mutants failed to rescue COX-II expression. The D136R and D139R mutants also failed to sustain growth in low-glucose conditions (Fig. 3F), whereas the D136A and D139A mutants showed no defects at 10 mM or 5 mM glucose, with only a mild reduction in D136A growth at 2 mM glucose. Collectively, these findings suggest that positive charge substitutions on D136 or D139 residues (and not the loss of the negative charge) on α-helix 3 inhibits both its ability to facilitate lipoylation and its function in cuproptosis.

## FDX1 mutants are enzymatically active in vitro

The D136 and D139 residues are solvent-exposed (Fig. 2C) and are in a region not predicted to significantly impact FDX1's overall structure. However, previous studies suggest that this evolutionarily conserved α-helix (Supplementary Fig. 2A) plays a key role in mediating interactions between FDX1 and both its upstream reductase FDXR and downstream substrate enzymes[5,6,19–23] (Fig. 4A). AlphaFold3[24] predictions of FDX1-FDXR binding indicate that D136 and D139 of FDX1 form a network of electrostatic interactions with R242, R271, and R275 of

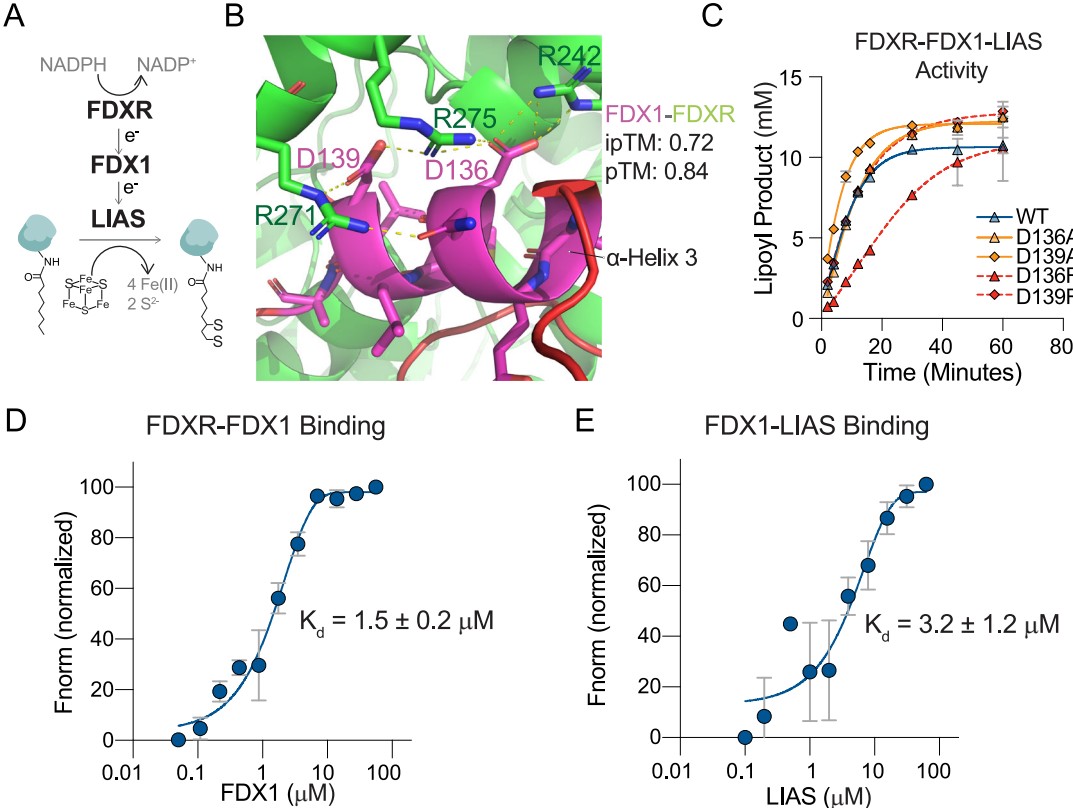

**Fig. 4 | D136 and D139 mutants are functional in vitro. A** Schematic of the canonical upstream regulation of FDX1. **B** AlphaFold3 prediction of the FDX1 (Uniprot: P10109) and FDXR (Uniprot: P22570) interaction. Shown in yellow dashed lines are the polar interactions (cutoff < 3.6 Å) on α-helix 3. (**C**) In vitro FDXR- and LIAS-mediated lipoylation activity assay that compares FDX1 WT versus the mutants. **D**, **E** In vitro TRIC fluorescence assay measuring the dissociation constant ($K_d$) between the indicated proteins. Data is presented as mean ± SD of three replicates from (**C**–**E**).

FDXR (Fig. 4B; Supplementary Fig. 2B, C). Consistent with this, similar electrostatic interactions are observed in the crystal structures of FDX1 and FDXR from *Bos taurus* PDB: 1E6E[20] and with an electron acceptor FNR from *Zea mays* PDB: 3W5U[25] (Supplementary Fig. 2D). In contrast, AlphaFold3's prediction of the binding between FDX1 double mutant DD136/139RR and FDXR fails to reach a confident score, highlighting the importance of the intricate electrostatic network between FDX1 α-helix 3 and FDXR (Supplementary Fig. 2E).

To determine the enzymatic activity of FDX1 mutants, we purified WT, D136, and D139 FDX1 mutants, where the residues were substituted with either an alanine (D136A, D139A) or an arginine (D136R, D139R). We then tested their ability to support LIAS-mediated lipoylation in-vitro. In this setting, FDX1 is reduced by FDXR (Fig. 4C). All FDX1 mutants retained activity in this FDX1-dependent lipoylation assay, although D136R exhibited slightly slower kinetics than the other variants. These results demonstrate that D136 and D139 mutations do not compromise FDX1's structural integrity and enzymatic potential.

To examine if these variations alter FDX1 protein-protein interactions, we employed Temperature-related Intensity Change (TRIC) fluorescence assays to measure FDX1 binding to both FDXR and LIAS. The results for FDX1 WT binding to both FDXR and LIAS are shown in Fig. 4D and Fig. 4E, respectively. The summarized results (Table 1) indicate that substituting the aspartic acids at these positions with either alanine or arginine does not significantly disrupt these interactions in vitro. Additionally, these mutations did not alter FDX1's midpoint potential (Table 1). While these findings establish that D136R and D139R are enzymatically active, they present a puzzling contradiction: despite retaining in vitro activity, these mutants fail to support lipoylation in cells. This discrepancy raises the possibility that, in human cells, an alternative upstream regulator distinct from FDXR interacts with FDX1 through electrostatic contacts that involve D136 and D139.

## DLD can replace FDXR in reducing FDX1

To explore whether additional upstream regulators of FDX1 exist (Fig. 5A), we used the Cancer Dependency Map (www.depmap.org) resource that measures the viability effect of genome-wide CRISPR/Cas9 loss-of-function gene knockouts in each individual cell line across more than a thousand different cell lines. We reasoned that the viability effect of genetically ablating *FDX1* should highly correlate with that of any upstream reductase or downstream substrates of FDX1. Using this gene dependency correlation analysis approach, we previously established that the primary role of FDX1 in human cells is to regulate mitochondrial protein lipoylation[12]. Surprisingly, correlating the *FDX1* gene dependency across DepMap does not show *FDXR* as a top correlating dependency, whereas the *FDX1* paralog *FDX2* does (Fig. 5A and Supplementary Fig. 3A). The top *FDX1* correlating dependencies were *LIAS* and *GCSH* (FDX1's downstream

**Table 1 | Summary of dissociation constants of the indicated paired proteins, as well as the midpoint potentials of wild type (WT) FDX1 or mutant forms**

|  | FDXR $K_d$ (µM) | LIAS $K_d$ (µM) | Midpoint potential (mV) |
|---|---|---|---|
| WT | 1.5 ± 0.2 | 3.2 ± 1.2 | −240 |
| D136A | 1.8 ± 0.3 | 3.2 ± 0.3 | −260 |
| D136R | 2.2 ± 0.3 | 4.5 ± 0.8 | −250 |
| D139A | 1.5 ± 0.3 | 4.9 ± 0.6 | −250 |
| D139R | 2.1 ± 0.2 | 2.5 ± 0.6 | −250 |

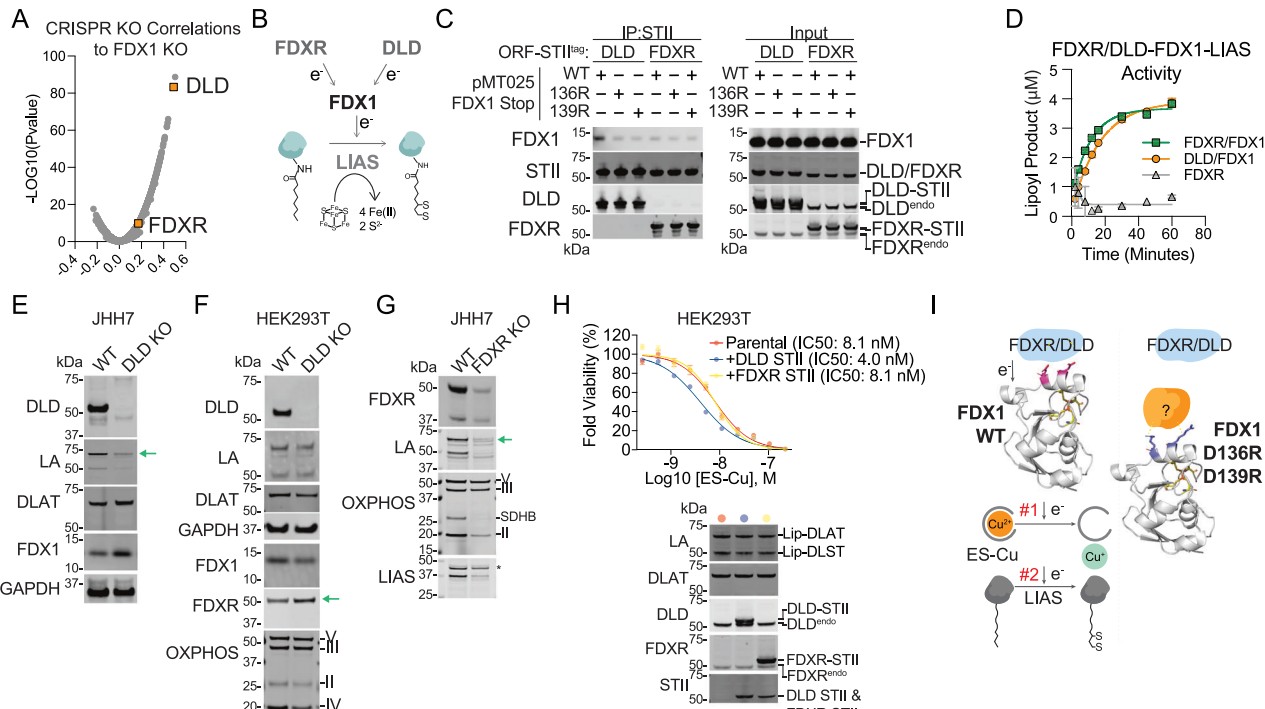

**Fig. 5 | DLD can serve as an alternative upstream regulator of FDX1. A** Volcano plot shows the CRISPR/Cas9 FDX1 loss-of-function viability outcome in 1373 cell lines correlated with the loss of function of all other tested genes. Plotted are the correlation values and the *p*-value for the 17,915 genes (see Methods for details). Possible FDX1 upstream regulators are colored orange. **B** Schematic of an alternative upstream regulation of FDX1. **C** Immunoprecipitation (IP) with anti-Strep-Tag II (STII) of C-terminally STII-tagged DLD or FDXR stably expressed in HEK293T cells. Indicated FDX1 constructs are transiently transfected. pMT025 is the same expression vector used for the DMS screen, and "stop" indicates a stop codon (i.e., no tag). ORF, open reading frame. Endo, endogenous. **D** In vitro FDX1- and LIAS-mediated lipoylation activity assay that compare FDXR and DLD as the upstream reducing source and FDXR alone as control. Data is presented as mean ± SD of three replicates. **E–G** CRISPR/Cas12a mediated *DLD* KO in JHH7 (**E**) and in HEK293T (**F**), and *FDXR* KO in JHH7 (**G**). The lysates were immunoblotted with the indicated antibodies. Green arrows highlight changes in levels of key proteins of interest. **H** Top, HEK293T overexpressing the indicated constructs were assayed for viability after 2 days of ES-Cu treatment. Bottom, immunoblots of cell lines shown above. Viability data is presented as mean ± SEM of five biological replicates. **I** Schematic depicting our model: during homeostasis, FDX1 uses its negative charges on its third alpha helix to facilitate critical downstream steps including lipoylation and Cu reduction in ES (left panel). By flipping the charges of those residues (right panel), an inhibitory protein binds mutant FDX1, preventing it from receiving electrons from DLD, FDXR, and/or other reducing sources. This results in a functionally inactive FDX1.

targets), along with dihydrolipoamide dehydrogenase (*DLD*), the E3 component in the lipoylated mitochondrial complexes. Although DLD's best-characterized enzymatic function is its dehydrogenase activity critical for the redox recycling of lipoic acid, it has also been shown to exhibit reductase (diaphorase) activity by oxidizing NADH in various contexts[26–35] (Supplementary Fig. 3B). This suggests that it could, in theory, serve as an upstream regulator of FDX1. Unexpectedly, unbiased structural homology analysis with DALI[36] and Foldseek[37], completed by targeted alignment using FATCAT[38], identified significant structural similarity between DLD and FDXR (Supplementary Fig. 3C, D), suggesting that DLD could be the functional upstream reductase of FDX1 in human cells (Fig. 5B).

To test this hypothesis, we used multiple complementary experimental approaches. First, we examined the interaction between FDX1 and DLD and compared it to that of FDX1 and FDXR. HEK293T cells were transfected with Strep-Tag (STII)-tagged DLD or FDXR constructs, followed by immunoprecipitation analysis for WT or mutant FDX1 binding. WT FDX1 showed substantially stronger binding with DLD than with FDXR. Notably, the D136R and D139R FDX1 mutants displayed decreased binding to DLD, while their interaction with FDXR remained unchanged (Fig. 5C). This pattern is consistent with our DMS results, which identified D136 and D139 as critical residues for FDX1 function, and suggested a specific role in mediating interactions with its upstream partner. The lack of differential binding of FDX1 and its mutants with FDXR further supports the idea that DLD, rather than FDXR, is the physiologically relevant

FDX1 reductase in this context. Next, we tested whether DLD could functionally replace FDXR in an in vitro LIAS-mediated lipoylation assay. DLD supported FDX1-dependent LIAS activity at a level comparable to that of FDXR (Fig. 5D). We then examined the effect of *DLD* gene knockout on protein lipoylation, reasoning that if DLD functions solely as the E3 component of lipoylated complexes, its loss should not alter total lipoylation levels. In JHH7 cells, *DLD* knockout lowered protein lipoylation levels by approximately four-fold when normalized to DLAT levels (Fig. 5E). The effect was more subtle in HEK293T (Fig. 5F), possibly reflecting a modest compensatory increase in FDXR levels. In contrast, *FDXR* deletion led to a loss of lipoylation as well as of Fe–S cluster proteins, including LIAS, making it difficult to determine whether FDXR supports lipoylation primarily by maintaining LIAS protein levels or through its FDX1–LIAS regulatory axis (Fig. 5G; Supplementary Fig. 3E). Finally, we wanted to test the relative contribution of either DLD or FDXR to FDX1-mediated cuproptosis. We found that in HEK293T DLD overexpression but not FDXR increased the sensitivity by approximately two-fold to ES-induced cuproptosis (Fig. 5H). Taken together, these results provide evidence that DLD can functionally serve as an upstream electron donor for FDX1. Collectively, these results support a model in which both DLD and FDXR contribute to FDX1 activation in cells, with their relative contributions likely depending on cell type and metabolic state.

Although FDX1 and DLD interact in cells, we did not detect direct binding between them in vitro using the TRIC assay (Supplementary

Fig. 3F, top panels). Moreover, FDX1 D136R and D139R mutants, which disrupted DLD binding in cells, retained full activity in the in vitro LIAS-mediated lipoylation assay with DLD as the reductant (Supplementary Fig. 3F, bottom panel). This apparent discrepancy between cellular and in vitro contexts suggests the involvement of an additional cellular factor that facilitates or stabilizes the FDX1–DLD interaction in cells but is absent in vitro. Despite this, our results clearly demonstrate that reversing negative charges on α-helix 3 of FDX1 disrupts its interaction with DLD in cells, correlating with severely abrogated ES-induced cuproptosis and a complete loss of protein lipoylation. These findings underscore the essential regulatory role of α-helix 3 and raise the possibility that DLD may function, at least in part, as the physiological reductase of FDX1 in cells (Fig. 5I).

## Discussion

Our study provides a comprehensive structure-function analysis of FDX1, suggesting that FDX1's dual roles in cuproptosis and mitochondrial protein lipoylation are structurally and functionally connected, with both processes requiring the same solvent-exposed residues on α-helix 3. Using deep mutational scanning, we identified D136 and D139 as essential for FDX1 activity in cells, where charge-reversal mutations abolished both Cu-induced toxicity and protein lipoylation. Despite this loss of function in cells, these mutants retained full enzymatic activity in vitro, revealing a context-dependent requirement for these residues. We further identify dihydrolipoamide dehydrogenase (DLD), the E3 component of lipoylated complexes as a previously unrecognized upstream reductase of FDX1. The differential impact of FDX1 D136R and D139R mutations on DLD versus FDXR binding, together with their functional outcomes in cells, suggests that DLD may mediate FDX1 reduction in cells through electrostatic interactions that are disrupted by these mutations. Collectively, these findings define a shared structural interface that links lipoylation and cuproptosis, and suggest that FDX1's physical and functional coupling to the lipoylation complex is the key determinant of cellular sensitivity to Cu-induced cell death.

α-helix 3 of FDX1 has been shown to be important in mediating FDX1 protein-protein interactions. However, these were not explored in the context of cuproptosis or lipoylation regulation. Crystal structures of the bovine FDXR–FDX1 complex show an acidic patch on FDX1 that docks onto complementary basic residues on FDXR. The negative charges of the three aspartic acids on α-helix 3 mediate the interactions of FDX1 with both FDXR and its downstream substrates such as the CYP protein family members[5,6,19–23]. These findings strongly indicate that α-helix 3 forms the crucial interface of FDX1 with both its upstream and downstream targets. However, this simplified model presents key gaps. If all FDX1 substrates interact via the same α-helix, additional mechanisms must regulate substrate specificity. This is particularly evident in the case of FDX1 and FDX2, which share high sequence homology yet exhibit distinct substrate preferences. Recent studies suggest that interactions with residue Glu133 and with the C-terminus play a crucial role in determining substrate specificity between these human ferredoxin paralogs[10]. However, if both FDX1 and FDX2 bind FDXR through the same α-helix 3 interface, it raises the possibility that alternative upstream regulators may exist, paralleling the differential substrate selection seen for their downstream targets.

The possibility of an alternative upstream regulator for FDX1 is further supported by gene dependency correlation analyses across hundreds of cancer cell lines (DepMap). Genes within the same pathway typically exhibit strong dependency correlations, as their knock-out effects on viability should be highly similar[12,39,40]. However, FDXR gene dependency correlates much more strongly with that of FDX2 than with FDX1, suggesting that FDXR primarily regulates FDX2, while FDX1 may rely on an alternative upstream reductase. Based on these observations, we nominated DLD as an alternative FDX1 reductase and demonstrated its role in regulating FDX1 activity. Nevertheless,

additional upstream regulators of FDX1 may exist in a context-dependent manner. One potential candidate is mitochondrial complex I, as previous genetic data indicate that its activity is required for cuproptosis induction[2]. Further studies will be needed to determine whether multiple mitochondrial reductases contribute to FDX1 regulation under specific cellular conditions.

DLD is best known to function as the E3 subunit of α-ketoacid dehydrogenase complexes, where it acts as a thiol dehydrogenase that transfers electrons from dihydrolipoamide to NAD$^+$ [26,39]. However, accumulating evidence suggests that DLD also possesses reductase activity with a variety of electron acceptors. Notably, DLD has been shown to catalyze one-electron reductions of chromium(VI) and vanadium(V)[28], Fe(III)[29], nitric oxide (NO)[32], ubiquinone[26], nitrofurans[27], and exhibits peroxidase activity[35]. The switch in DLD enzymatic function appears to be regulated by structural changes affecting its oligomerization: while DLD dimers and tetramers retain both lipoamide dehydrogenase and diaphorase activities, the monomeric form exhibits only diaphorase activity[31]. This transition is further influenced by pH[31], substrate availability[35], and metal ions, particularly zinc[30,33]. Although no direct evidence currently links Cu regulation to DLD activity, its metal-dependent enzymatic modulation raises the intriguing possibility that Cu itself may regulate DLD function.

The profound inhibitory effect of the charge-flip mutations on all tested FDX1 functions in cells suggests that these residues are critical for interactions with an upstream regulator. This is supported by our finding that DLD binds directly to FDX1 in cells, and that this interaction is severely impaired by the D136R and D139R mutations. However, this is difficult to reconcile with the mutants' preserved activity in vitro, where they retain full enzymatic function, maintain binding to FDXR and LIAS, and support DLD-mediated LIAS activity. Moreover, while DLD efficiently reduces FDX1 in vitro, a property consistent with the reduction potentials of DLD (E1: −346 mV, E2: −280 mV)[41] and FDX1 (−240 mV), strong FDX1–DLD binding is observed only in cells. One model that could explain this discrepancy is that, in cells, the charge-flip mutants aberrantly engage an inhibitory factor which is absent in vitro and prevents FDX1 reduction by upstream partners (Fig. 5I). Alternatively, additional regulatory mechanisms such as post-translational modifications, conformational changes, oligomerization states, or adapter proteins may be required to facilitate WT FDX1 binding to upstream reductases in the cellular environment. In sum, the fact that the same mutations simultaneously abolish both FDX1-mediated lipoylation and cuproptosis further supports the conclusion that these functions are mechanistically linked. We propose that, in cells, FDX1 operates within a larger multiprotein complex that governs mitochondrial protein lipoylation, and that the assembly of this complex is essential for enabling FDX1-dependent lipoylation and the induction of cuproptosis.

## Methods

### Cell line generation

Cell lines were created as previously described[12]. The ribonucleoprotein (RNP) used in FDX1 KO by nucleofection was formed using a 1:1 ratio of Alt-R CRISPR-Cas9 tracrRNA (100 μM) and Alt-R CRISPR-Cas9 predesigned crRNA (100 μM) (Hs.Cas9.FDX1.1.AD IDT guide) (Integrated DNA Technologies). The mixture was incubated at 95 °C for 5 min and then brought to room temperature to form the gRNA duplex. The gRNA duplex (50 μM) was then combined with Alt-R Cas9 enzyme (61 μM) (Integrated DNA Technologies) at a ratio of 3:2 and incubated at room temperature for 10−20 min to form the RNP that was used for the nucleofection described below. Nucleofection of ABC1 (from the Genetic Perturbation Platform of the Broad Institute) and HEK293T cells (from ATCC CRL-3216) was conducted according to the manufacturer's protocol (Lonza). 500,000 cells per sample were pelleted by centrifugation at 4 °C using a tabletop centrifuge at 800 x g for 5 min. Cell culture media supernatant was

aspirated and discarded, and the cells were resuspended in 30 μL of a mixture containing 20 μL Lonza SF cell line solution, 5 μL FDX1 Cas9 RNP, 1 μL electroporation enhancer, and 4 μL PBS (Lonza Biosciences). Each cell suspension was added to individual wells of an electroporation cuvette (Lonza Biosciences), and then nucleofected using the Lonza 4D system program A549 pulse code CM-130. Cells were then transferred from the cuvette into pre-warmed media in a 6-well or 12-well culture plate and allowed to grow to 80% confluence. The nucleofected HEK 293 T cells were further single-cell plated to generate FDX1 KO single-cell clones. ABC1 FDX1 KO cells could not be single cell cloned, and the bulk nucleofected population was validated for high-efficiency KO and used in subsequent experiments. For the generation of the FDX1 KO cell lines with reconstituted FDX1, pMT025 FDX1 vector (see below) was used to make lentiviral particles. These particles were produced in HEK293T cells, and the FDX1 KO cells (ABC1, HEK293T) were infected and selected with puromycin to ensure homogenous FDX1 gene expression.

### Viability assays (Cell-Titer Glo and Cellcyte)
Cell-Titer Glo (CTG) assays are performed as per manufacturer's protocol. Briefly, ABC1 and HEK293T cells were plated in 384 well plates and left to attach overnight. Drugs were then treated as indicated. After the indicated incubation time, CTG reagent was added at equal volume as the media volume. For cell proliferation assays, cells stably expressing histone 2 GFP (H2GFP), which enable accurate live cell number quantification, were plated at 5000–10,000 cells per well in 96-well plates. Then, the plates were imaged with Cellcyte Live Cell Analyzer (Cytena) at 6-h intervals for three to 6 days. Image analysis was done using the Cellcyte analysis software, quantifying each GFP+ nuclei as one cell. Proliferation rates are plotted in $\log_2$ fold change (Fig. 3B, C) from initial count or as confluence % over time (Fig. 3F).

### Immunoprecipitation (IP)
HEK293T stably expressing either DLD or FDXR are transfected with the indicated FDX1 constructs (1 ug of DNA) in six-well plates. After 2 days of transient transfection, cells were harvested and lysed in PBS containing 1% IGEPAL and HALT protease and phosphatase inhibitors. Lysates were then clarified at 10,000 x g for 10 min at 4 C; this serves as the input for the IP. Strep-Tactin Magnetic beads were then added to each sample, and the mix was incubated in the cold for 30 min rotating at low speed. Beads were then washed with the lysis buffer, incubated with 1x SDS loading dye (with reducing reagent), and boiled at 95 °C for 10 min. Samples were then separated by SDS-PAGE and immunoblotted with the indicated antibodies.

### Structural Modeling
Individual AlphaFold (AF) structures were retrieved from the Alpha-Fold Database (DB), which was generated by the AF Monomer v2.0 piepline and was last updated on July 31st, 2025. Specifically, for FDX1 the model used is AF-P10109-F1-V6. For DLD the model used is AF-P09622-F1-V6. For FDXR the model used is AF-P22570-F1-V6. The predicted interaction between FDX1 and FDXR was generated by Alphfold3, and was performed on the AlphaFold server (alpha-foldserver.com). The disordered amino acid sequences were trimmed from the input (as indicated on Supplementary Fig. 2C), as we reasoned that their high flexibility could bias the model toward non-specific contacts and depress confidence metrics. The model was generated using the default options, and the resulting.pdb files were inspected and visualized using Pymol. DALI and Foldseek are programs that offer structure alignments against large protein structure collections. DALI server analysis (ekhidna2.biocenter.helsinki.fi/dali/) was performed using the AF-DB comparison search, and the Foldseek server analysis (search.foldseek.com/search) was restricted to

mammalian structures only. Flexible structure AlignmenT by Chaining Aligned fragment pairs allowing Twists (FATCAT)'s pairwise alignment function was used to align and analyze the FDXR and DLD structural similarity. Models were visualized using Pymol.

### Cloning of pMT025 FDX1
Full length FDX1 (NM_004109.5) was cloned into the pMT025 backbone vector (Addgene plasmid #158579) previously described[17,42]. Briefly pMT025 vector was digested using NheI and BamHI and FDX1 ORF inserted using Gibson assembly protocol.

### Design and generation of FDX1 saturation mutagenesis library
The FDX1 has 183 amino acids excluding the start and stop codons. For each amino acid position of the FDX1 ORF reference sequence, either 20 or 21 alternative codons were selected, one for each of the 19 possible amino acid substitutions, one for a nonsense mutation, and, in some cases, also a codon corresponding to a silent mutation. Whenever multiple choices of codon were available, we selected codons that differed from the reference sequence by 2 or 3 nucleotides rather than codons for which only 1 nucleotide differed from the reference sequence. This was done to reduce confounding contributions to variant read counts due to errors in PCR amplification or sequencing miscalls, both of which are predominantly single-nucleotide (detailed in ref. 42). Silent mutations were included only for the 81 residues for which silent mutations could be achieved with a 2- or 3-nucleotide change to the reference sequence.

In all, the FDX1 variant library consists of 3740 variants, 20 alterations at each of the 183 amino acid positions, plus 81 silent alleles, minus one missense variant that was excluded because there was no codon option that avoided conflict with the cloning sites. A library of FDX1 full-length ORF variant oligonucleotides was synthesized at Twist BioScience. The ORF sequences were flanked with adapters for cloning. The fragment library was digested with NheI and BamHI, and ligated into the pMT_025 vector (available from Addgene). The ligated construct was used to transform Stbl4 bacterial cells and plasmid DNA (pDNA) was extracted using QIAGEN Maxi Prep Kits. The resulting pooled ORF variant plasmid DNA library was sequenced via Illumina Nextera XT platform to determine the distribution of variants within the library.

### DMS screen in HEK293T and ABC1 cells
Lentivirus was produced by transfecting 293T packaging cells with TransIT-LT1 (Mirus Bio) and three plasmids: the pooled FDX1 ORF pDNA library, psPAX2 (Addgene), and pMD2.G (Addgene). Medium was replaced 6–8 h post-transfection, and viral supernatant was collected ~24 h later. After titration on each target line, $3 \times 10^6$ HEK293T and ABC1 cells were infected at an MOI of 0.3. Following selection and expansion, $1.0 \times 10^7$ cells were allocated per experimental arm.

### FDX1 DMS PCR amplification and deconvolution
The general screen deconvolution strategy and considerations were described in detail in ref. 42. The integrated open reading frame (ORF) in genomic DNA was amplified by PCR. The PCR products were shotgun sheared with transposon, index labeled, and sequenced with next-generation sequencing technology. The PCR primers were designed in such a way that there is a ~100 bp extra sequence at each end leading up to the mutated ORF region. We used 2 primers: (Forward: 5'-ATTCTCCTTGGAATTTGCCCTT-3'; Reverse: 5'-CATAGCGTAAAAGGAG CAACA-3'). PCR reactions were performed for each gDNA sample with a reaction volume of 50 uL and with 1 ug gDNA. Q5 (New England BioLabs) was used as DNA polymerase. 1/3 of 96 PCR reactions of a gDNA sample were pooled, concentrated with Qiagen PCR cleanup kit, and then purified by 1% agarose gel. The excised bands were purified first by Qiagen Qiaquick kits, then by AMPure XP kit (Beckman Coulter). Following Illumina Nextera XT protocol, for each sample, we set

up 6 Nextera reactions, each with 1 ng of purified ORF DNA. Each reaction was indexed with unique i7/i5 index pairs. After the limited-cycle PCR step, the Nextera reactions were purified with AMPure XP kit. All samples were then pooled and sequenced with Illumina Nova-seq S4 platform. NovaSeq600 S4 data were processed with software AnalyzeSaturationMutagenesis, ASMv1.0 for short, which was developed by Broad Institute as previously described[40]. Typically, the pair-end reads were aligned to the ORF reference sequence. Multiple filters were applied, and some reads were trimmed. The counts of detected variants were then tallied. The output files from ASMv1.0, one for each screening sample, were then parsed, annotated merged into a single.csv file that is ready for hit-calling utilizing software tools that are freely available as previously described[40].

### DMS analysis
Abundance of each variant was calculated by the fraction of reads compared to the total reads of all variants in each endpoint, and $\log_2$ Fold Change (LFC) was determined between elesclomol treatment compared to DMSO control. There were two replicates per treatment group, and the duplicates were averaged for analyses. To better appreciate our variant activity relative to wildtype FDX1, the DMS LFC was centered to the mean of the wildtype FDX1 (silent mutations) distribution and normalized against the mean of FDX1 non-sense variants.

### In silico saturation mutagenesis with FoldX
The in silico saturation mutagenesis studies on apo-FDX1 structure PDB: 3P1M that evaluate the protein stability from the perspective of free energy change ($\Delta\Delta G$) upon mutations were performed using the FoldX[16]. MutateX[43] was used for automation to systematically mutate each available residue to all other possible residues. The RepairPDB function of FoldX was first applied for energy minimization to modify the protein system to reasonable conformations. The BuildModel function was followed for the computational mutagenesis and reporting $\Delta\Delta G$ values.

### Protein expression and purification
Recombinant protein purification: the protein coding sequence of DLD (aa 36-509), FDX1 WT (aa 61 – 184), FDXR (a.a 33–491) and LIAS (aa 28–372) all lacking their respective mitochondrial transit peptides were custom synthesized at Thermo Fisher Scientific after codon optimization for recombinant protein overexpression in E. coli. The DLD, FDXR and LIAS genes were subcloned into a modified pSUMO vector (LifeSensors Inc.), (pDWSUMO), while the FDX1 WT and its D136A, D136R, D139A, and D139R mutant genes were subcloned into pET28a vector. The genes were cloned using NdeI and XhoI restriction sites of the appropriate vector. After gene sequence verification at Penn State Genomics Core Facility (University Park, PA), the DLD and FDXR plasmids were separately transformed into E. coli BL21 (DE3) cells, while LIAS, FDX1 WT, and its variants were separately transformed into BL21 (DE3) cells carrying pDB1282, the plasmid which harbors an isc operon from A. vinelandii[44]. DLD, FDXR, and LIAS were each overexpressed as a fusion with a ULP1 protease cleavable N-terminal SUMO tag that also carried a hexa-histidine tag at its N-terminus, while FDX1 and variants were overproduced with an N-terminal hexa-histidine tag. During purification, the SUMO tag was cleaved from the fusion protein, affording pure respective proteins with a Gly-His appendage at the N-terminus. All proteins were over-expressed and purified, with the respective iron-sulfur clusters (only for FDX1 and LIAS) reconstituted following procedures as previously published for LIAS with no further modifications[45].

### In vitro FDX1 dependent lipoylation assays
All in vitro activity assays were conducted as previously reported[45]. Assays were conducted in an anaerobic chamber. In each assay, the following components were added at their final concentrations (unless otherwise noted) in a final volume enough for all time points as needed: buffer (50 mM HEPES, pH 7.5, 250 mM KCl) containing 10 μM LIAS, 10 μM SAH nucleosidase, 15 μM FDX1, 5 μM FDXR (or DLD), 1 mM NADPH, 0.75 mM SAM and 350 μM octanoyllysyl-containing peptide substrate (Glu-($N^6$-octanoyl) Lys-Ala-Tyr). The reactions were incubated at 30 °C for 10 min before initiating with SAM, quenched at specific time points, and analyzed as previously reported[45].

### Temperature-Related Intensity Change (TRIC) Fluorescence Assays
TRIC is a biophysical technique that detects binding events between a fluorescently labeled target and a ligand by measuring fluorescence intensity changes induced by localized temperature shifts. The interactions between FDX1 and LIAS, and FDXR and DLD were measured via the TRIC method. For these experiments, a Dianthus NT.23 instrument (NanoTemper Technologies GmbH, München, Germany) was used. For the binding assays, RED-NHS 2nd generation protein labeling kit (NanoTemper Technologies GmbH) was used to label tagless FDXR for its binding measurements to N-terminus 6His-tagged FDX1 WT and its variants. Alternatively, Red-Tris-NTA 2nd generation protein labeling kit (NanoTemper Technologies GmbH) was used to label N-terminus 6His-tagged FDX1 WT and its variants for binding measurements to tagless LIAS. All the labeling and binding steps of the experiment were performed in an anaerobic chamber (Coy Laboratory products, Grass Lake, Michigan). The labeling, binding and data analysis steps were carried out following the procedures as recommended by the manufacturer of the instrument and the labeling kits. The TRIC measurements were performed in triplicate at room temperature in TRIC Buffer (25 mM MOPS, pH 7.5, 125 mM KCl, 125 mM NaCl, 5% glycerol, 0.05% Tween). For FDXR-FDX1 assays, 50 nM final concentration of labeled FDXR target was used while the FDX1 ligand was serially diluted (each with 11-point serial dilutions from: 56 μM (WT); 61 μM (D136A); 63 μM (D136R); 68 μM (D139A); and 54 μM (D139R)). For FDX1-LIAS assays, 100 nM final concentration of labeled FDX1 WT and its variants as target was used while LIAS was serially diluted (with 10-point serial dilutions in each case from 63 μM) as the ligand. The FDXR-FDX1 and LIAS-FDX1 mixtures were incubated for 30 min at room temperature. TRIC data was collected using the DI. control data collection software (NanoTemper Technologies GmbH) at 22 °C using default settings with laser on-time set to 5 s with auto-excitation. The data was plotted using GraphPad Prism v8.3.0.

### Redox potential measurements
Electrochemical experiments were carried out anaerobically in a Coy glovebox using a WaveNow wireless potentiostat (Pine). A three-electrode configuration was used in a water jacketed glass cell. A platinum wire was used as the counter electrode and the reference electrode was a standard Ag:AgCl electrode; potential reported are relative to the normal hydrogen electrode. Baseline measurements were collected using a pyrolytic graphite edge (PGE) electrode polished with 600 μm alumina slurry and applied 10 uL of 2.5 mg/mL multi-walled carbon nanotube (MWCNT) in dimethylformamide (DMF). The PGE:MWCNT was rinsed and placed into a glass cell containing a RT mixed buffer solution (10 mM MES, CHES, TAPS, HEPES), pH 6.5 with 300 mM NaCl. A 5 μL aliquot of 1.5 mM protein with 13 mg/mL poly-myxin B sulfate Salt (Sigma) solution co-adsorbent was applied directly to the polished PGE:MWCNT electrode surface and incubated for 4 mins. The excess protein sample was removed, and the electrode was placed back into the buffer cell solution. Non-turnover electro-chemical signals were analyzed by correction of the non-Faradaic component of the current from the raw data using the QSOAS package.

### Immunoblot Analysis.
Cells were lysed using RIPA lysis buffer (Sigma-Aldrich) with protease inhibitor cocktail tablets (Sigma-Aldrich) and

Article

incubated on ice for 30 min. Samples were subjected to centrifugation for 10 min at 10,000 x g to pellet cell debris. The soluble fraction was collected for quantification, and the insoluble fraction was discarded. Protein was quantified using the bicinchoninic acid (BCA) method with a bovine-specific albumin standard curve for normalization. Each protein extract was added to final 1x LDS sample buffer with a 1:10 TCEP solution reducing agent and size fractionated via pre-cast SDS-PAGE Bis-Tris 4 to 12% gels (Thermo Fisher Scientific). Gels were then transferred onto nitrocellulose membranes with the transblot turbo system (BioRad). Each membrane was then incubated at room temperature for 1 h in LICOR Odyssey blocking buffer, followed by overnight incubation at 4 °C with the indicated antibody. The membranes were then washed four times with 1× TBS with 0.1% Tween20 (TBST) and incubated with fluorophore-specific IRDye secondary antibodies (LI-COR) at 1:10,000 for an hour. The membranes were then washed again four times with TBST before being imaged on a LI-COR Odyssey machine. Antibodies used in this study include (all at 1:1000): FDX1 (Abcam ab108257), Lipoic acid (Millipore, cat# 437695 and Abcam ab58724), LIAS (Proteintech 11577-1-AP), DLAT (Cell signaling 12362S and Thermo), DLST (Cell signaling 5556S), OXPHOS (Abcam ab110411), FDXR (Proteintech 15584-1-AP), DLD (Proteintech 16431-1-AP), and Strep Tag II (STII) (Invitrogen MA5-37747). All relevant antibodies were previously validated for specificity with knockouts.

## Correlation analysis
CRISPR *FDX1* dependency scores across 1128 cell lines from the Broad Institute's public 24Q4 dependency map were correlated with other CRISPR dependency scores from the Broad Institute's 24Q4 Dependency Map/CCLE release. Pearson's correlations were performed in R using the cor.test function. *P*-values were then corrected for false discovery using the Benjamini-Hochberg method of the p.adjust function in R and q-values were -log10-normalized. Top 1000 correlating genes were plotted using GraphPad Prism v8.3.0.

## Reporting summary
Further information on research design is available in the Nature Portfolio Reporting Summary linked to this article.

## Data availability
All data associated with the DMS screen are available in on the data repository FigShare [https://doi.org/10.6084/m9.figshare.30606473.v1]. All other processed data and uncropped immunoblots are available in the source data. The solved crystal structures used in this study can be accessed from the following PDB codes: 1E6E, 3W5U, 3P1M. The accession codes of the structures retrieved from the AlphaFold Database are indicated in the methods section. All additional structural predictions using AlphaFold3 and structural alignment using FATCAT are provided in the source data. Original fastq files were uploaded to NCBI SRA database with BioProject ID PRJNA1373349. Source data are provided with this paper.

## Code availability
All relevant code to analyze the DMS screen is available on the following GitHub link: https://github.com/jkwonbio/FDX1.

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

## Acknowledgements

This work was supported by the National Cancer Institute grant R01CA279550 (P.T.), National Institutes of Health (awards GM-122595 to S.J.B.), the National Science Foundation (MCB-1716686 to S.J.B.), and the Eberly Family Distinguished Chair in Science (to S.J.B.). S.J.B. is an investigator of the Howard Hughes Medical Institute. The content is solely the responsibility of the authors and does not necessarily represent the official views of the National Institutes of Health.

## Author contributions

P.T. conceptualization. J.C.H., and P.T. performed the cell-based experiments. D.M.W., performed and analyzed the in vitro activity and binding experiments with the help of T.B-L. M.B.D. and N.R.B. established cell line models and helped set up DMS screen. J.J.K. analyzed the DMS screen. D.E.R. devised the DMS experimental setup and library cloning. S.J.B. analyzed and interpreted the in vitro studies. P.T. analyzed and interpreted the results. P. T. and J.C.H. wrote the manuscript, which was read and approved by all authors.

## Competing interests

The authors declare no competing interests.
