## [Transparent Peer Review file · Nature Communications]

Deep Mutational Scanning of FDX1 Identifies Key Structural Determinants of Lipoylation and Cuproptosis

Corresponding Author: Dr Peter Tsvetkov

Version 0:

Reviewer comments:

Reviewer #1

(Remarks to the Author)

Recent studies have highlighted critical roles of mitochondrial reductase – the Ferredoxin 1 (FDX1) in diverse biochemical processes, including steroid hormone synthesis, lipoylation, electron transport chain complex IV biogenesis, as well as copper-mediated cell death. In this context, this study by Hsiao et al., to elucidate the molecular basis of FDX1 function is an important advance. Here, the authors used deep mutational scanning to identify two key amino acids residues, D136 and D139 that are critical for mediating FDX1 function in both cuproptosis and protein lipoylation. Interestingly, despite losing function in cells, these FDX1 mutants maintained full enzymatic activity in vitro when FDXR was used as the electron donor. This suggests that FDX1 may receive electron from another donor in cells. Using data from the Cancer Dependency Map and follow-up experiments, the authors identified dihydrolipoamide dehydrogenase (DLD) as a likely upstream reductase that activates FDX1 in vivo. Firmly establishing DLD's role in FDX1 activation will greatly strengthen the paper.

Comments:

1. Are the redox potentials of FDX1 and DLD consistent with the proposed electron donor role of DLD?
2. Since copper-transporting property of elesclomol has been shown to be evolutionarily conserved, are aspartate residues (D136 and D139) also conserved?
3. Some discrepancy between the in vivo and in vitro data regarding DLD-FDX1 functional interaction (Figure 5C and 5E) prevents reaching clear conclusions about the electron donor role of DLD.
4. There is a discrepancy in kd values depicted in Figure 4D and 4F. In one case, it is reported as 1.5 +/- 1.2 (Figure 4D) and other case it is reported as 1.5 +/- 0.2 (Figure 4F). This typographical error should be corrected.

Reviewer #2

(Remarks to the Author)

Reviewer #3

(Remarks to the Author)

The paper entitled "Deep Mutational Scanning of FDX1 Identifies Key Structural Determinants of Lipoylation and Cuproptosis" by Hsiao and collaborators describes the critical role of two aspartic acid residues present in the α -helix 3 of mitochondrial FDX1 for supporting lipoylation and cuproptosis in cells. Charge-reversal mutations of these residues, initially selected via a deep mutational scanning approach, do not affect FDX1 structure or enzymatic activity as assessed by in vitro experiments but have important cellular consequences on the above-mentioned processes. It is concluded that these two functionally important residues are required for favoring FDX1 reduction in cells by the dihydrolipoamide dehydrogenase (DLD) rather than by the ferredoxin reductase.

Overall, this is an original and well executed study and the results support the conclusions. I only have a few concerns about the description of some methods and results that are detailed below.

1. Cell viability tests after treatment with elesclomol were sometimes performed after 72 hours, as shown in Figures 1B and 3 but also after 14 and 11 days for data shown in Figure 1B. Is there a rationale for this difference? By the way, the text (lines 76-82) is written such that we understand that two cell models (HEK293T and ABC1) have been analyzed but results shown in Fig 1B are those obtained using ABC1 cells. Add results about HEK293T cells in the main figure or as supplemental figure?

2. There is a need to better describe the cell lines used and to present all control lines and data.

In Figure 1: One cell line used in the study is a FDX1 KO overexpressing FDX1. It may be important to indicate to which extent the cells overexpress FDX1. It seems that the same cells have been used for immunoblots shown in Figure 3 which should allow providing quantitative data and discussing whether indeed the levels of cell viability correlates with expression levels. If this is correct, then the naming is confusing since they are referred to as KO mutant complemented with WT FDX1 in Figure 3 (and no more as FDX1 KO + FDX1 OE). Please clarify this point. On this aspect, the "methods" section also requires improvement as it is only indicated in lines 411-412 that FDX1 was cloned into the pMT025 backbone vector". What was cloned exactly? the full-length gene? which promoter and terminator were used?

In Figure 3B/C why is there no KO + WT as a reference here while they are used for other experiments?

In Figure 3E, why is there no parental line as a reference here?

3. in line 140-141, there is a statement "Defects in the integrity of the ETC Complex IV were similarly observed in these cells" only referring to the reference 13 but no indication of a related figure. Do we have to understand that the immunoblots shown in Figures 3D and 3E for COXII CIV support this statement? If yes, the data obtained for ABC1 and HEK293T cells are not so comparable. Some explanation/clarification is needed here.

4. The authors mentioned in the discussion that the activity profile of DLD differs according to the oligomerization forms which can be monomers, dimers/tetramers (lines 272-274). Because several experiments have been performed using DLD (enzymatic and interaction assays, Fig 5D, S3E), it seems important to this reviewer to provide indications about the homogeneity of the sample used for the experiments.

Minor comments:

1. Line 78: the term "transient electroporation" does not seem appropriate. Transient expression?

2. Line 134-135: The following sentence "To assess the impact of D136 and D139 substitutions on protein lipoylation, we expressed WT and mutant FDX1 in FDX1 KO ABC1 and HEK293T cells." may be rephrased since this information is provided just above (lines 124-129).

3. Line 156: The use of a maize FDX-FNR complex for structural comparison is maybe not appropriate, first because these are chloroplastic proteins and second because FNR is the electron acceptor in this case and not the electron donor.

4. Line 185: sentence that needs a correction "Using this approach, we were previously able to determine that the primary role of..."

5. Line 486: Recombinant protein purification - Genes encoding DLD (36-509), FDX1 WT (aa 61 – 184)... Change genes by coding sequences since there may be introns in the genes?

6. Line 544: Define these acronyms: MWCNT in DMF.

7. In Figure 2, replace Fe-S binding by Fe-S cluster binding.

Reviewer #4

(Remarks to the Author)

Hsiao et al present a comprehensive study of FDX1's role in cuproptosis. They explore FDX1's two main roles in reducing copper ions and regulating protein lipoylation. They use an innovative Deep Mutational Scanning (DMS) approach to explore the impact of every possible FDX1 variant on overall protein function, as revealed by a cell viability assay. Their analyses (DMS but also a computational stability analysis) revealed two conserved residues, D136 and D139, in FDX1's α -helix 3 as essential for lipoylation and cuproptosis, though mutations at these sites preserve enzymatic activity in vitro. They further explore these two residues and several other aspects of FDX1 function (outside our main area of expertise which is DMS). The core DMS experiments are well-performed and shed insight into FDX1 function, and we applaud the authors on using this method to gain insight into protein function. However, several important methods details are not presented and some of the conclusions are somewhat overstated.

Major comments:

The first sentence of the discussion: "...FDX1's dual roles in cuproptosis and mitochondrial protein lipoylation are structurally and functionally inseparable" is overstated. The authors validated several FDX1 residues as being important for both roles, but it is possible that there are other mutants that do separate these two functions that were not explored in the paper. The authors main DMS assay is a general cell viability assay that does not distinguish between cuproptosis and lipoylation (which is fine—but don't overstate conclusions).

The authors focus on positions D136 and D139 for further detailed analysis. These two positions come from a combination of the DMS and a computational stability analysis. There are several other interesting "hotspot" regions in the DMS map that

go unmentioned or barely mentioned in the paper, including regions near residues 40, 70, 134, and 152. (It is hard to tell precisely because there is no numbering of positions in the heatmap, which should be added). Although it is not necessary for the authors to perform in-depth followup experiments on these residues as with 136+139, can the authors at least discuss these other DMS hotspots, in regards to structural elements/location in protein/hypotheses for mechanism? Other structure-function conclusions like the fact that nonsense variants post ~170 are not loss of function go undiscussed as well.

It is important to note that the stability metric is a computational prediction, not an experimental measurement (which could have been experimentally measured by a VAMP-seq DMS approach). We would recommend changing "Higher Global Protein Instability" in Figure 2B-C to "Higher Predicted Protein Instability" and similar edits in the text. A caveat to the singular focus on 136+139 is that other variants could have predicted loss-of-stability but actually have normal stability.

Some DMS methods are missing or unclear. How was the DMS mutant library generated? How was it delivered into cells? How many cells were grown/transfected/harvested?

General minor comments:

Add "MAVE" to keywords

In addition to Figure 1A, it would be helpful to have a broader overview figure drawn of the cuproptosis pathway/other proteins/pic of mitochondria, etc that is the main subject of the introduction+paper.

More details on FDX1 knockout needed. Sanger confirmed? Homozygous or heterozygous?

How many total mutations were DMS scores were obtained for and what fraction of possible mutants?

Spell out HEK293 and ABC1 at first use. It is a strength that the DMS experiment was performed in both cell lines with similar results, but it is unclear why the 2 cell lines were chosen or necessary. Was there a difference or hypothesized difference between the two lines in terms of cuproptosis or FDX1 function?

Re: "The effects of FDX1 mutations on ES sensitivity were highly correlated between the two cell lines." Can the authors plot the scores from the two cell lines against each other and mention the correlation coefficient in the results?

How many DMS replicates were performed in each cell line? What was the correlation of scores between the replicates?

How were replicates (if any) handled in the analysis?

Figure 2 minor comments:

Add numbers to the heatmap indicating residue numbers

Why isn't deldelG calculated for the N-terminus?

Suggest to label residue types in Figure 1A (e.g. hydrophobic, polar, charged etc)

There is excellent discrimination between nonsense and silent mutants, indicating that the assay+data quality is excellent.

However, consider scaling the heatmap colors more gradually so that there is more white on the heatmap because the silent mutations had a bit of a spread of scores.

Were late nonsense mutants (with normal scores) excluded in the violin plot of nonsense mutants, and excluded for the normalization of scores?

Methods minor comments:

Refers to "viral" particles without saying what type of virus.

Was every variant included in the DMS analysis or were there QC criteria for excluding a variant (for example based on low abundance in the initial sample).

Data availability:

The standard in the DMS/MAVE community is to deposit scores to MAVE-DB before or at time of publication. In addition, depositing raw high-throughput sequence data in NCBI Sequence Read Archive before or at time of publication. I would request these from the authors.

Reviewer #5

(Remarks to the Author)

Version 1:

Reviewer comments:

Reviewer #1

(Remarks to the Author)

In the revised manuscript, the authors have tried to address my concern regarding the discrepancy between the in vivo and in vitro data by presenting new results in the rebuttal letter. However, these new data on DLD-KO and FDXR-KD (Rebuttal letter Figure 2) are not included in the revised manuscript. I recommend incorporating these data into the manuscript to strengthen the conclusions. The authors should note that the figure legend for this new data (Rebuttal Figure 2) does not match the figure and will need to be corrected. In the same figure, the authors should also include the sensitivity of DLD knockout to ES-Cu in Panel C. These new data will significantly bolster the proposed model and merit publication in this high-impact journal.

Reviewer #2

(Remarks to the Author)

Reviewer #3

(Remarks to the Author)

In this revised version of the manuscript, the authors have adequately addressed my previous concerns.

Reviewer #4

(Remarks to the Author)

The authors satisfactorily answered my questions.

Reviewer #5

(Remarks to the Author)

We thank the editor and reviewers for their constructive feedback, which substantially improved the manuscript. In response, we (i) clarified the central conclusions and tempered language where appropriate; (ii) added new in-cell experiments to probe the role of DLD in FDX1 activation, including DLD knockouts/overexpression across two cell models and ES–Cu sensitivity assays; (iii) corrected typographical/consistency issues; (iv) expanded Methods; and (v) revised figures/legends. We also now discuss additional DMS hotspots and caveats regarding predicted versus measured stability. We believe these changes address each comment and strengthen the mechanistic framework. Please see detailed response below.

Our point-to-point responses are colored in blue,
Figure, citations, and line callouts are bolded and highlighted in yellow,
Recited lines are quoted and italicized,
Figure legends are colored in dark blue.

Reviewer #1 (Remarks to the Author):

Recent studies have highlighted critical roles of mitochondrial reductase – the Ferredoxin 1 (FDX1) in diverse biochemical processes, including steroid hormone synthesis, lipoylation, electron transport chain complex IV biogenesis, as well as copper-mediated cell death. In this context, this study by Hsiao et al., to elucidate the molecular basis of FDX1 function is an important advance. Here, the authors used deep mutational scanning to identify two key amino acids residues, D136 and D139 that are critical for mediating FDX1 function in both cuproptosis and protein lipoylation. Interestingly, despite losing function in cells, these FDX1 mutants maintained full enzymatic activity in vitro when FDXR was used as the electron donor. This suggests that FDX1 may receive electron from another donor in cells. Using data from the Cancer Dependency Map and follow-up experiments, the authors identified dihydrolipoamide dehydrogenase (DLD) as a likely upstream reductase that activates FDX1 in vivo. Firmly establishing DLD's role in FDX1 activation will greatly strengthen the paper.

We thank the reviewer for succinctly summarizing the paper's main point and for the constructive feedback. As described in detail below, we have made a concerted effort to address the question of DLD's role in FDX1 activation in cells through a series of cell-based experiments.

Comments:

1. Are the redox potentials of FDX1 and DLD consistent with the proposed electron donor role of DLD?

Yes, the redox potentials of FDX1 and DLD are consistent with the proposed role of DLD as an electron donor to FDX1. Specifically, midpoint potentials at pH 7.0 are as follows: DLD E2 (enzyme disulfide) **-0.280 V**; DLD E1 (flavin) **-0.346 V** (**Matthews & Williams Jr., 1976**); FDX1 **-0.240 V** (**Fig. 4F**). Thus, electron transfer from DLD to FDX1 is thermodynamically favorable under standard conditions. We have now emphasized this point in the text (**lines 368 to 370**):

*“Moreover, while DLD efficiently reduces FDX1 in vitro, a property consistent with the reduction potentials of DLD (E1: -346 mV, E2: -280 mV) (**PMID: 6467**) and FDX1 (-240 mV), strong FDX1–DLD binding is observed only in cells.”*

2. Since copper-transporting property of elesclomol has been shown to be evolutionarily conserved, are aspartate residues (D136 and D139) also conserved?

Yes, these two aspartic acid residues are conserved across evolution, extending to bacterial ferredoxins. Please refer to **Rebuttal Fig. 1** below (and the now updated **Fig. S2A**), which includes an amino acid sequence alignment across multiple species. We have also added this Figure call-out at **lines 202-204**.

Rebuttal Figure 1: Aspartic acids 136 and 139 are evolutionarily conserved. Multiple sequence alignment (using Clustal Omega) of the regions surrounding the third alpha helix in FDX1 from the indicated species. Asterisk, complete conservation; colon, strong conservation; period, weak conservation. The two completely conserved aspartic acids of interest (human D136 and D139) are highlighted in light red, and human E135 in light blue. The region that constitutes α -Helix 3 is underscored.

3. Some discrepancy between the *in vivo* and *in vitro* data regarding DLD-FDX1 functional interaction (Figure 5C and 5E) prevents reaching clear conclusions about the electron donor role of DLD.

We agree with the reviewer that there is a discrepancy between our *in vivo* and *in vitro* data regarding the functional interaction between DLD and FDX1, which we have acknowledged in both the Results and Discussion sections of the manuscript. It is important for us to further emphasize that the discrepancy between the *in vivo* and *in vitro* data is true for both the D136R and D139R mutants, as they show no lipoylation in cells. Yet, they retain their function and binding to both FDXR and DLD in the *in vitro* setting. This point is re-emphasized in our discussion (**lines 366 – 382**). In this rebuttal, we focused our attention to reaching clearer conclusions about the reductase activity of DLD by conducting a series of *in cell* experiments, which we deem the more physiologically relevant context.

To recap, our experiments thus far showed that the strong binding of FDX1 to DLD in cells is disrupted upon the introduction of the D136R or D139R mutations (**Fig. 5C**). This finding along with others allowed us to hypothesize that DLD may be the predominant upstream regulator of FDX1 in cells. To test this hypothesis, we generated DLD knockout (KO) lines in two cellular models, HEK293T and JHH7, which we have previously used to study protein lipoylation (**PMID: 37453661**).

If DLD's sole role were an E3 subunit of lipoylated complexes, DLD loss would not be expected to affect overall protein lipoylation. However, we observed a cell line specific loss of lipoylation:

an approximately four-fold reduction of lipoylated protein levels in JHH7 when normalized to total DLAT levels, but no loss in HEK293T cells (**Rebuttal Fig. 2A-B**). We do, however, see a modest increase in FDXR levels in HEK293T, perhaps suggesting compensatory mechanisms. Next, we wanted to test the relative contribution of either DLD or FDXR to FDX1-mediated cuproptosis. We found that in HEK293T, DLD overexpression but not FDXR increased the sensitivity by ~2 fold to elesclomol-induced cuproptosis (**Rebuttal Fig. 2C**). Given that global lipoylation levels remain unchanged in the DLD overexpression cell line, this supports its direct functional impact on cuproptosis by acting on FDX1. Moreover, given that DLD drives dihydrolipoamide to lipoamide, DLD overexpression is expected to deplete lipoyl thiolates, making it unlikely that it sensitizes cells to cuproptosis by increasing copper-reactive lipoyl thiolate targets. Collectively, these results support a model in which both DLD and FDXR contribute to FDX1 activation in cells, with their relative contributions likely depending on cell type and metabolic state.

We also considered the reciprocal approach of knocking out FDXR. However, this experiment cannot resolve the specific contribution to FDX1 activation, as FDXR loss also disrupts Fe-S cluster biosynthesis, leading to loss of LIAS (and SDHB) and thus an indirect decrease in global lipoylation levels (**Rebuttal Fig. 2D**). This confounding variable therefore cannot allow us to establish FDXR's role in reducing FDX1 in cells.

We believe that these results have allowed us to reach clearer conclusions about the electron role of DLD, specifically in the more physiologically relevant in-cell context. We have emphasized this point more explicitly in the revised Discussion (**lines 364 - 382**).

“This is supported by our finding that DLD binds directly to FDX1 in cells, and that this interaction is severely impaired by the D136R and D139R mutations. However, this is difficult to reconcile with the mutants’ preserved activity in vitro, where they retain full enzymatic function, maintain binding to FDXR and LIAS, and support DLD-mediated LIAS activity. Moreover, while DLD efficiently reduces FDX1 in vitro... strong FDX1–DLD binding is observed only in cells. One model that could explain this discrepancy is that, in cells, the charge-flip mutants aberrantly engage an inhibitory factor which is absent in vitro and prevents FDX1 reduction by upstream partners (Fig. 5F). Alternatively, additional regulatory mechanisms such as post-translational modifications, conformational changes, or adaptor proteins may be required to facilitate WT FDX1 binding to upstream reductases in the cellular environment.

Rebuttal Figure 2: DLD and FDXR contribute to FDX1 activation in a context-dependent manner. (A-C) DLD was knocked out in HEK293T (A) and in JHH7 (B), while FDXR was knocked out in JHH7 (C). The lysates were immunoblotted with the indicated antibodies. (D) Top, three HEK293T cell lines of wildtype parental, stable overexpression of STII-tagged DLD, and stable overexpression of STII-tagged FDXR were assessed for their sensitivities to elesclomol-Cu induced cuproptosis, and the IC50 values are indicated. Bottom, the lysates were

immunoblotted with the indicated antibodies. (E) Proposed model: both DLD and FDXR contribute to FDX1 activation, but FDXR also indirectly regulates lipoic acid synthesis by maintaining proper levels of iron-sulfur cluster biogenesis.

4. There is a discrepancy in kd values depicted in Figure 4D and 4F. In one case, it is reported as 1.5 +/- 1.2 (Figure 4D) and other case it is reported as 1.5 +/- 0.2 (Figure 4F). This typographical error should be corrected.

We thank the reviewers for catching the typographical error. We have now fixed it to the correct value of 1.5 +/- 0.2 μ M in the updated **Figure 4D**.

Reviewer #2 (Remarks to the Author):

Reviewer #3 (Remarks to the Author):

The paper entitled “Deep Mutational Scanning of FDX1 Identifies Key Structural Determinants of Lipoylation and Cuproptosis” by Hsiao and collaborators describes the critical role of two aspartic acid residues present in the α -helix 3 of mitochondrial FDX1 for supporting lipoylation and cuproptosis in cells. Charge-reversal mutations of these residues, initially selected via a deep mutational scanning approach, do not affect FDX1 structure or enzymatic activity as assessed by in vitro experiments but have important cellular consequences on the above-mentioned processes. It is concluded that these two functionally important residues are required for favoring FDX1 reduction in cells by the dihydrolipoamide dehydrogenase (DLD) rather than by the ferredoxin reductase. Overall, this is an original and well executed study and the results support the conclusions. I only have a few concerns about the description of some methods and results that are detailed below.

We thank the reviewer for their positive evaluation of our study. We appreciate the constructive comments, which we address point-by-point below.

1. Cell viability tests after treatment with elesclomol were sometimes performed after 72 hours, as shown in Figures 1B and 3 but also after 14 and 11 days for data shown in Figure 1B. Is there a rationale for this difference? By the way, the text (lines 76-82) is written such that we understand that two cell models (HEK293T and ABC1) have been analyzed but results shown in Fig 1B are those obtained using ABC1 cells. Add results about HEK293T cells in the main figure or as supplemental figure?

The rationale for the extended 11- and 14-day treatments in the DMS screen is to ensure robust selection of cells resistant to elesclomol-induced cuproptosis. This longer exposure period is standard in most screening workflows, as it maximizes the likelihood of identifying truly resistant clones while minimizing noise. In contrast, all individual mutant validations are performed over 72 hours, during which we consistently observe the reported phenotypes.

Regarding the calibration of our system in HEK293T, we have now added the relevant data to **Fig. S1A (Rebuttal Fig. 3)**, and have revised the text to call out the figure in **lines 84 to 87**:

“As expected, knockout (KO) of FDX1 conferred resistance to ES-induced cuproptosis, whereas reconstitution of FDX1 in KO cells re-sensitized them, yielding even greater sensitivity than in the wild-type control (~100-fold increase) in the ABC1 cell line (Fig. 1B; S1A).”

Rebuttal Figure 3: Elesclomol (ES)-induced cuproptosis is dependent on FDX1 in HEK293T. Top, viabilities were assessed by Cell Titer Glo in HEK293T WT, FDX1 KO, and FDX1 KO complemented with FDX1 WT. Bottom, immunoblots of the cell lines.

2. There is a need to better describe the cell lines used and to present all control lines and data. In Figure 1: One cell line used in the study is a FDX1 KO overexpressing FDX1. It may be important to indicate to which extent the cells overexpress FDX1.

We have now included immunoblots directly comparing FDX1 levels in the complemented cell line to those in the parental line (Fig. S1A; Rebuttal Fig. 3). The results show significant increase in FDX1 levels in the FDX1 KO + WT FDX1 cell line over the parental. To avoid confusion, we have now standardized the nomenclature throughout the manuscript to refer to this model consistently as “FDX1 KO complemented with FDX1 WT.” The revised text are as follows:

Line 164 to 166: “These mutants, along with wild-type (WT) FDX1, were complemented to similar levels in FDX1 knockout (KO) ABC1 and HEK293T cells, and their sensitivity to ES or ES-Cu was assessed (Fig. 3A-C).”

Line 177 to 187: “Complementation of WT, D136A, or D139A mutants completely rescued lipoylation, suggesting that these aspartic acid residues are dispensable and can be substituted by other amino acids without an overall loss of enzymatic function.”

It seems that the same cells have been used for immunoblots shown in Figure 3 which should allow providing quantitative data and discussing whether indeed the levels of cell viability correlates with expression levels. If this is correct, then the naming is confusing since they are referred to as KO mutant complemented with WT FDX1 in Figure 3 (and no more as FDX1 KO + FDX1 OE). Please clarify this point.

Because the relative FDX1 levels are relatively similar as indicated in **Fig. 3D** among the complemented cell lines (KO +WT or +mutants), the levels of cell viability are principally dependent on the biological effects of FDX1's ability to mediate elesclomol-induced or glucose deprivation-induced toxicity.

On this aspect, the “methods” section also requires improvement as it is only indicated in lines 411-412 that FDX1 was cloned into the pMT025 backbone vector”. What was cloned exactly? the full-length gene? which promoter and terminator were used?

We have expanded the Methods section to specify that the full-length human FDX1 coding sequence (Uniprot ID: P10109) was cloned into the pMT025 lentiviral backbone under the control of the EF1 α promoter, with the SV40 polyadenylation signal used as the terminator right after a WPRE signal (**lines 467 to 469**). We will be depositing these plasmids onto Addgene.

In Figure 3B/C why is there no KO + WT as a reference here while they are used for other experiments?

We have now included KO + WT as a reference in **Fig. 3B**. While performing the experiment, we did not include the KO + WT cell line in **Fig. 3C**, as we reasoned that the marked differential sensitivity of the alanine mutants to the arginine mutants was sufficient to emphasize the point that the former could mediate elesclomol-induced cuproptosis but not the latter.

In Figure 3E, why is there no parental line as a reference here?

Fig. S1A (Rebuttal Fig. 3) now show immunoblots directly comparing FDX1 levels in the complemented cell line to those in the parental line.

3. in line 140-141, there is a statement “Defects in the integrity of the ETC Complex IV were similarly observed in these cells” only referring to the reference 13 but no indication of a related figure. Do we have to understand that the immunoblots shown in Figures 3D and 3E for COXII CIV support this statement? If yes, the data obtained for ABC1 and HEK293T cells are not so comparable. Some explanation/clarification is needed here.

We thank the reviewer for pointing out the lack of Figure call-out in the text and for pointing out the discrepancy between the data obtained from ABC1 and HEK293T. We agree that the extent of the defect appears much more pronounced in HEK293T cells compared to ABC1, which we have not further investigated here as we believe it is a phenomenon beyond the scope of this manuscript. We, however, speculate the difference may reflect intrinsic variations in how FDX1

contributes to the heme a/a₃ synthesis pathway (as proposed by reference 13) between the two cell lines. We have now clarified this difference in the revised text and explicitly linked the statement to the relevant figure (lines 188 to 193).

“Previously, FDX1 was shown to regulate biogenesis of mitochondrial cytochrome c oxidase in human cells¹³. The effect of FDX1 KO on COX-II levels seems to be cell line specific, with dramatic loss of COX-II in HEK293T but not ABC1 FDX1 KO cells (Fig. 3D-E). In HEK293T cells, expression of wild-type FDX1 as well as the D136A and D139A mutants successfully restored COX-II levels, whereas the D136R and D139R mutants failed to rescue COX-II expression.”

4. The authors mentioned in the discussion that the activity profile of DLD differs according to the oligomerization forms which can be monomers, dimers/tetramers (lines 272-274). Because several experiments have been performed using DLD (enzymatic and interaction assays, Fig 5D, S3E), it seems important to this reviewer to provide indications about the homogeneity of the sample used for the experiments.

By using Blue-Native PAGE analysis, we have assessed the oligomerization state of the DLD used to perform the in vitro enzymatic and binding assays. The results show that DLD exists homogeneously as a dimer, as shown below in Rebuttal Fig. 4. We believe that the dimeric form is the active species that donates electrons to FDX1.

Rebuttal Figure 4: Bacterial-purified human DLD migrates as a dimer. Lane identities are as indicated on the Blue-Native PAGE analysis. Expected molecular weight of dimeric DLD is 104 kDa.

Minor comments:

1. Line 78: the term “transient electroporation” does not seem appropriate. Transient expression?

We thank for reviewer for pointing out the confusing terminology. We have now fixed it by deleting the word transient to describe the electroporation process (lines 81 to 83):

“First, we established two cell line models (ABC1 and human embryonic kidney 293 cells T expressing SV40 large T-antigen [HEK293T]) in which FDX1 was deleted using electroporation of Cas9 ribonucleoprotein complexes, avoiding constitutive Cas9 overexpression.”

2. Line 134-135: The following sentence “To assess the impact of D136 and D139 substitutions on protein lipoylation, we expressed WT and mutant FDX1 in FDX1 KO ABC1 and HEK293T cells.” may be rephrased since this information is provided just above (lines 124-129).

We agree with this suggestion, and we have now fixed the text to avoid redundancy (lines 174 to 175):

“FDX1 facilitates cuproptosis by reducing Cu(II)-ES to Cu(I) and regulating protein lipoylation^{1-3,10-12}. Therefore, we further assessed the impact of D136 and D139 substitutions on protein lipoylation levels in ABC1 and HEK293T cells.”

3. Line 156: The use of a maize FDX-FNR complex for structural comparison is maybe not appropriate, first because these are chloroplastic proteins and second because FNR is the electron acceptor in this case and not the electron donor.

We thank for the reviewer for pointing this out, which we acknowledge we had previously missed. Considering FDX1’s conservation in prokaryotes (Rebuttal Fig. 1), we still believe that plant FDX1 is a relevant comparison due to several reasons. First, the acidic alpha helix is still observed in plant FDX1 (residues 117 to 123 in maize FDX1 [Uniprot P27787]), which demonstrates structural homology to human FDX1. Second, human FDX1 also uses the same acidic alpha helix to interface with the electron acceptor Cytochrome P450 enzymes. Lastly, plant FDX1, which contain an iron-sulfur cluster, participates in similar electron transfer reactions within plastids, providing a mechanistic parallel that supports its utility as a comparative model. That said, we agree with the reviewer that differences in subcellular localization, physiological context, and potential partner proteins between plant and human systems represent important caveats to this comparison. We have revised the text in the manuscript accordingly (lines 202 to 204).

“However, previous studies suggest that this evolutionarily conserved α -helix plays a key role in mediating interactions between FDX1 and both its upstream reductase FDXR and downstream substrate enzymes^{5,6,18-22} (Fig. 4A, Fig. S2B)... electrostatic interactions are observed in the crystal structures of FDX1 and FDXR from bovine (PDB: 1E6E)¹⁹ and maize (PDB: 3W5U)²⁴ (Fig. S2A).”

4. Line 185: sentence that needs a correction “Using this approach, we were previously able to determine that the primary role of...”

We have now corrected this sentence (line 256 to 257) to ensure clarity.

“Using this gene dependency correlation analysis approach, we previously revealed and established that the primary role of FDX1 in human cells is to regulate mitochondrial protein lipoylation¹².”

5. Line 486: Recombinant protein purification - Genes encoding DLD (36-509), FDX1 WT (aa 61 – 184)... Change genes by coding sequences since there may be introns in the genes?

We have now corrected this sentence (line 561 to 564) to ensure clarity.

“Recombinant protein purification: the protein coding sequence of DLD (aa 36-509), FDX1 WT (aa 61 – 184), FDXR (a.a 33–491) and LIAS (aa 28–372) all lacking their respective mitochondrial

transit peptides were custom synthesized at Thermo Fisher Scientific after codon optimization for recombinant protein overexpression in E. coli.”

6. Line 544: Define these acronyms: MWCNT in DMF.

We have now defined these acronyms in the text (lines 625 to 627).

“Baseline measurements were collected using a pyrolytic graphite edge (PGE) electrode polished with 600 μ m alumina slurry and applied 10 μ L of 2.5 mg/mL multi-walled carbon nanotube (MWCNT) in dimethylformamide (DMF).”

7. In Figure 2, replace Fe-S binding by Fe-S cluster binding.

We have now updated Fig. 2A and Fig. S1B.

Reviewer #4 (Remarks to the Author):

Hsiao et al present a comprehensive study of FDX1’s role in cuproptosis. They explore FDX1’s two main roles in reducing copper ions and regulating protein lipoylation. They use an innovative Deep Mutational Scanning (DMS) approach to explore the impact of every possible FDX1 variant on overall protein function, as revealed by a cell viability assay. Their analyses (DMS but also a computational stability analysis) revealed two conserved residues, D136 and D139, in FDX1’s α -helix 3 as essential for lipoylation and cuproptosis, though mutations at these sites preserve enzymatic activity in vitro. They further explore these two residues and several other aspects of FDX1 function(outside our main area of expertise which is DMS). The core DMS experiments are well-performed and shed insight into FDX1 function, and we applaud the authors on using this method to gain insight into protein function. However, several important methods details are not presented and some of the conclusions are somewhat overstated.

We thank the reviewer for their positive evaluation of our work and for recognizing the value of our DMS approach in uncovering FDX1 function. We appreciate the constructive feedback regarding methodological details and the interpretation of our conclusions, and we have addressed these points in detail below.

Major comments:

The first sentence of the discussion: “...FDX1’s dual roles in cuproptosis and mitochondrial protein lipoylation are structurally and functionally inseparable” is overstated. The authors validated several FDX1 residues as being important for both roles, but it is possible that there are other

mutants that do separate these two functions that were not explored in the paper. The authors main DMS assay is a general cell viability assay that does not distinguish between cuproptosis and lipoylation (which is fine—but don't overstate conclusions).

We agree that our current data do not exclude the possibility of mutants that could separate FDX1's roles in cuproptosis and lipoylation. We have revised the first sentence of the discussion to remove the overstatement and to clarify that our DMS assay measures overall cell viability, which may reflect contributions from both processes.

Lines 29 to 31: *“Together, these findings establish α -helix 3 of FDX1 as a critical interface for its upstream regulation and suggest that FDX1's roles in lipoylation and cuproptosis are both structurally and functionally linked.”*

Lines 303 to 306: *“Our study provides a comprehensive structure-function analysis of FDX1, suggesting that FDX1's dual roles in cuproptosis and mitochondrial protein lipoylation are structurally and functionally connected...”*

The authors focus on positions D136 and D139 for further detailed analysis. These two positions come from a combination of the DMS and a computational stability analysis. There are several other interesting “hotspot” regions in the DMS map that go unmentioned or barely mentioned in the paper, including regions near residues 40, 70, 134, and 152. (It is hard to tell precisely because there is no numbering of positions in the heatmap, which should be added). Although it is not necessary for the authors to perform in-depth follow up experiments on these residues as with 136+139, can the authors at least discuss these other DMS hotspots, in regards to structural elements/location in protein/hypotheses for mechanism? Other structure-function conclusions like the fact that nonsense variants post ~170 are not loss of function go undiscussed as well.

We thank the reviewer for highlighting these additional mutational hotspots in our data. In response, we have added a new section discussing these regions, including their structural context and possible functional relevance, with guidance from structural modeling and in silico analysis (**Figure S1C-F**). In the interest of space, we have chosen not to reproduce the figures here in this rebuttal, and we apologize for any inconvenience. All comments below are incorporated in the text (**lines 103 to 130**). We have also added numbering of positions in the heatmap.

For residues near 40 to 50, AlphaFold predicts the first 65 amino acids as disordered, and the solved crystal structure (PDB 3P1M) lacks this region. We hypothesize that this region serves as a mitochondrial localization sequence. Indeed, when we performed in silico analysis on sequentially truncated versions of FDX1, the score for FDX1's mitochondrial localization drops sharply when it is truncated at position 41 from the N-terminus (**Fig. S1C**). Collectively, these indicate the possibility that residues 40 to 50 form part of the mitochondrial targeting sequence, and that mutations in this region may impair mitochondrial import and thus indirectly affect FDX1's function of cuproptosis.

For residues V69 and F71, we show by structural modelling that they help define a hydrophobic core with beta-strand character in this region (**Fig. S1D**). We hypothesize that their mutations significantly disrupt the fold of FDX1 and thus impair its enzymatic activity.

For residue E134, structural modelling indicates that it coordinates the side chain of R149 and the backbone of L149 (Fig. S1E). This glutamate residue is in fact highly conserved, down to bacterial FDX1 (Fig. S2B). We hypothesize that this residue is critical for stabilizing the acidic alpha helix in which D136 and D139 reside.

As for residue C152, it coordinates the iron-sulfur cluster on FDX1 and is therefore essential for FDX1's enzymatic activity (Fig. 2C).

For residues after 167, we hypothesize that since AlphaFold predicts that the region starting from position 167 is disordered, the extreme C-terminal tail is not absolutely required for a functional FDX1 (Fig. S1F).

It is important to note that the stability metric is a computational prediction, not an experimental measurement (which could have been experimentally measured by a VAMP-seq DMS approach). We would recommend changing “Higher Global Protein Instability” in Figure 2B-C to “Higher Predicted Protein Instability” and similar edits in the text. A caveat to the singular focus on 136+139 is that other variants could have predicted loss-of-stability but actually have normal stability.

We have updated Fig. 2B-C and the corresponding text to use the term “Higher Predicted Protein Instability,” emphasizing that the stability metric is computationally derived. We have also added a statement acknowledging that predicted loss of stability may not always correspond to experimentally measured instability across all variants and have cited VAMP-seq-based DMS as a potential validation approach (lines 136 to 139).

“Loss-of-function (LOF) variants predicted to cause global protein instability exhibit high $\Delta\Delta G$ values. It should be noted that this metric represents a computational prediction rather than an experimental measurement, and may therefore result in false positives/negatives (PMID: 29785012).”

Some DMS methods are missing or unclear. How was the DMS mutant library generated? How was it delivered into cells? How many cells were grown/transfected/harvested?

We have incorporated the necessary details into the text (lines 482 to 500).

A smaller excerpt of the methods section is reproduced here:

“In all, the FDX1 variant library consists of 3740 variants, 20 alterations at each of the 183 amino acid positions, plus 81 silent alleles, minus one missense variant that was excluded because there was no codon option that avoided conflict with the cloning sites. A library of FDX1 full-length ORF variant oligonucleotides was synthesized at Twist BioScience. The ORF sequences were flanked with adapters for cloning... 293T viral packaging cells were transfected using TransIT-LT1 transfection reagent (Mirus Bio) with three plasmids: the pooled ORF pDNA library, a lentiviral packaging plasmid containing gag, pol and rev genes (psPAX2, Addgene), and an envelope plasmid containing VSV-G (pMD2.G, Addgene)....”

DMS screen in HEK293T and ABC1 cells. *Lentivirus was produced by transfecting 293T packaging cells with TransIT-LT1 (Mirus Bio) and three plasmids: the pooled FDX1 ORF pDNA library, psPAX2 (Addgene), and pMD2.G (Addgene). Medium was replaced 6–8 h post-transfection, and viral supernatant was collected ~24 h later. After titration on each target line,*

3x10⁶ HEK293T and ABC1 cells were infected at an MOI of 0.3. Following selection and expansion, 1.0x10⁷ cells were allocated per experimental arm. ”

General minor comments:

Add “MAVE” to keywords

We have now added MAVE to keywords.

In addition to Figure 1A, it would be helpful to have a broader overview figure drawn of the cuproptosis pathway/other proteins/pic of mitochondria, etc that is the main subject of the introduction+paper.

We agree with this suggestion, and we have now added a schematic depicting our current molecular model of the cuproptosis pathway into Fig. 1A.

More details on FDX1 knockout needed. Sanger confirmed? Homozygous or heterozygous?

Although we have not confirmed the KOs by Sanger sequencing, our immunoblots of the knockout consistently show a significant reduction in FDX1 protein levels (by one order of magnitude in ABC1s and below the detection limit in HEK293Ts), as indicated in Fig. 3D & S1A. Accordingly, we were able to achieve a complete or near-complete functional loss of FDX1 in both models, consistent with the reduction of FDX1 levels as detected by immunoblotting. We also note that these same knockout cell lines have been extensively characterized in our previous study establishing FDX1 as a critical regulator of de novo lipoic acid synthesis (PMID: 37453661, 35298263).

How many total mutations were DMS scores were obtained for and what fraction of possible mutants?

Scores were obtained for 3,740 variants (missense = 3470; nonsense = 183; silent = 81). Excluding the first methionine, FDX1 contains 183 amino acids that can be mutated to 19 other amino acids to produce 3,477 missense mutations. Our variant library of missense mutations covers 99.8% of all possible variants. We have now outlined these details in Methods Section (lines 482 to 500).

Spell out HEK293 and ABC1 at first use. It is a strength that the DMS experiment was performed in both cell lines with similar results, but it is unclear why the 2 cell lines were chosen or necessary. Was there a difference or hypothesized difference between the two lines in terms of cuproptosis or FDX1 function?

We have revised the text to spell out HEK293T at its first use, but ABC1 does not appear to be an acronym (lines 81 to 83).

The ABC1 cell line was selected because it was identified in a Broad Institute-developed PRISM (Profiling Relative Inhibition Simultaneously in Mixtures) screen of over 700 cell lines as one of

the most sensitive to elesclomol-induced cuproptosis (PMID: 31133756). HEK293T cells were included as an independent model to test the generalizability of hits from our DMS screen across different cellular contexts. As such, the DMS screen performed with ABC1 resulted in generally higher signal-to-noise ratio, but the same hits of interest were similarly observed in the HEK293Ts. It should be emphasized that both cell lines were characterized previously for their FDX1 dependent regulation of protein lipoylation (PMID: 37453661, 35298263).

Re: “The effects of FDX1 mutations on ES sensitivity were highly correlated between the two cell lines.” Can the authors plot the scores from the two cell lines against each other and mention the correlation coefficient in the results?

We appreciate the insightful suggestion. We have now incorporated a linear model between the two screens in Fig. S1H (Rebuttal Fig. 5). We find an overall linear regression model coefficient of $R = 0.46$, with a highly significant Pearson correlation p-value (< 0.001). We have now included these details within the Results section (lines 102-104).

Rebuttal Figure 5: Scatterplot indicates each FDX1 mutant’s LFC in viability in HEK293Ts (y-axis) and in ABC1s (x-axis). Mutations of interest are indicated. LFC, log fold change. The correlation coefficient and the p-value is indicated. Colored in maroon are the positively charged substitutions of D136/139. Colored in pink are the other substitutions of D136/139, and colored in green are the nonsense mutations of all positions.

How many DMS replicates were performed in each cell line? What was the correlation of scores between the replicates? How were replicates (if any) handled in the analysis?

There were two replicates per treatment condition. Except for the replicates of the elesclomol treatment arm in HEK293T, each replicate had good corollary with one another between replicates (Rebuttal Fig. 6). As described in Methods (lines 545-550), replicates were averaged together for the analyses.

Rebuttal Figure 6: Scatterplot indicates each FDX1 mutant’s LFC in viability in HEK293Ts (y-axis) and in ABC1s (x-axis). Mutations of interest are indicated. LFC, log fold change. The correlation coefficient and the p-value is indicated. Colored in maroon are the positively charged substitutions of D136/139. Colored in pink are the other substitutions of D136/139, and colored in green are the nonsense mutations of all positions.

Why isn’t del Δ G calculated for the N-terminus?

We did not calculate structure-based $\Delta\Delta$ G values for the N-terminal region because the first amino acids are absent from the available FDX1 crystal structure (PDB 3P1M) and they are predicted to be disordered and function as the mitochondrial targeting signal. Structure-based $\Delta\Delta$ G methods, including the FoldX method we used here (Methods Section) requires a pdb file to base these free energy calculations. We have clarified this rationale in the revised text.

Suggest to label residue types in Figure 1A (e.g. hydrophobic, polar, charged etc)

We have now added annotations for biophysical properties of each amino acid substitution to the heatmap to indicate residue numbers.

There is excellent discrimination between nonsense and silent mutants, indicating that the assay+data quality is excellent. However, consider scaling the heatmap colors more gradually so that there is more white on the heatmap because the silent mutations had a bit of a spread of scores.

We thank the reviewer for this constructive suggestion. The heatmap has been re-scaled to provide a more gradual color gradient, increasing the proportion of neutral (white) values to better visualize score variation among silent mutations. This adjustment improves resolution of subtle differences while preserving the clear discrimination observed between nonsense and silent variants, supporting the overall high assay performance and data quality.

Were late nonsense mutants (with normal scores) excluded in the violin plot of nonsense mutants, and excluded for the normalization of scores?

We appreciate the reviewer’s thoughtful question regarding the treatment of late nonsense mutations. We carefully considered whether to exclude C-terminal nonsense variants that

retained near-normal activity. However, we found the statistical basis for drawing an objective cutoff to be challenging. Several late nonsense positions exhibited partial or semi-loss-of-function behavior, making it difficult to consistently exclude some while retaining others without introducing subjective bias. For this reason, and after deliberate evaluation, we opted not to remove these variants from the violin plot or from the normalization process. We believe this approach avoids arbitrary exclusions and, if anything, renders our analysis more stringent, as it holds all nonsense variants to the same standard rather than selectively filtering.

Methods minor comments:

Refers to “viral” particles without saying what type of virus.

We have fixed this lack of clarification (line 428-429). Lentivirus was used.

Was every variant included in the DMS analysis or were there QC criteria for excluding a variant (for example based on low abundance in the initial sample).

All variants were initially included in the DMS analysis. We carefully evaluated whether to impose quality-control thresholds based on plasmid DNA (pDNA) abundance or Day 0 representation. Specifically, we examined the relationship between plasmid counts and log fold-change (LFC) values, as well as the distribution of Day 0 counts across replicates. Our analyses (see code) showed that variants with lower plasmid representation (e.g., ≤ 50 counts) did not systematically bias LFC measurements, and there was no clear correlation between initial abundance and assay outcome. Accordingly, we chose not to impose an arbitrary abundance cutoff, other than removing variants where log fold-change values were undefined (e.g., $-\infty$ when the denominator equaled zero). Variants with zero Day 0 counts or missing data at endpoints were set to NA and excluded from downstream calculations by necessity, but otherwise no additional filtering was applied. This approach maximizes transparency and inclusivity of the dataset while still ensuring stringency by discarding only values that were mathematically undefined.

Data availability:

The standard in the DMS/MAVE community is to deposit scores to MAVE-DB before or at time of publication. In addition, depositing raw high-throughput sequence data in NCBI Sequence Read Archive before or at time of publication. I would request these from the authors.

We agree with the reviewer that deposition of both processed variant scores and raw high-throughput sequencing data is an important standard for ensuring reproducibility and broad accessibility in the DMS/MAVE community. We are working on organizing our data so that we can deposit our complete variant effect score dataset in MAVE- and submit the raw FASTQ files to the NCBI Sequence Read Archive before the paper is published. Both datasets will be publicly available at the time of publication, with accompanying documentation describing file structure, processing pipelines, and relevant metadata to facilitate reuse.

Reviewer #5 (Remarks to the Author):

We thank the reviewers for their helpful comments and suggestions. Following Reviewer #1's recommendation, we have added new panels (Fig. 5E–G) and updated the manuscript text accordingly (lines 254–270). Also all raw FASTQ files have been fully deposited to NCBI (PRJNA1373349), and all processed source data are available through Figshare (<https://doi.org/10.6084/m9.figshare.30606473.v1>), and Github (<https://github.com/jkwonbio/FDX1>) these repositories provide everything necessary for full reproducibility and reanalysis. We will try our best to complete the MAVE upload within the next month. However, the MAVE deposition process is not intuitive and requires substantial time and effort, and we do not currently have the personnel or technical bandwidth to prioritize this. Given that all mandated data have already been deposited and are publicly accessible, we do not think that publication should be contingent on the MAVE accession. If the journal or reviewers can provide direct guidance or assistance with the MAVE upload, we would truly appreciate it and be happy to coordinate.

We also implemented the editorial revisions requested throughout the manuscript to improve clarity and accuracy.